# ONLINE LEARNING OF NONLINEAR AUTOREGRESSIVE PROCESSES OVER CELLULAR COMPLEXES

## ABSTRACT

Real-world time-series are often high-dimensional, structured with higher-order dependencies, and exhibit time-varying dynamics, making them challenging to model effectively. Abstract Cellular Complexes (ACCs) provide a principled way to capture such higher-order topological structure, serving as a powerful inductive bias for multivariate time-series modeling. While recent methods based on message-passing neural networks and Hodge Laplacians exploit higher-order relationships, they are primarily designed for offline settings and lack adaptability to streaming and non-stationary environments. In this work, we introduce a framework for nonlinear autoregressive modeling over ACCs, where predictive functions are defined in a reproducing kernel Hilbert space (RKHS) induced by shift-invariant kernels. We further propose an efficient online learning algorithm to estimate these functions. Experimental results on synthetic and real-world datasets demonstrate that our method shows competitive or improved performance in prediction accuracy and adaptability under streaming conditions.

## 1 INTRODUCTION

Modeling multivariate time-series is challenging due to high dimensionality, nonlinear dependencies, and dynamic behavior commonly observed in real-world systems Lütkepohl (2005); Tank et al. (2021); Zaman et al. (2021). In many such systems, the temporal evolution of signals is closely tied to an underlying higher-order structure Isufi et al. (2025). For instance, in water networks, pressures at junctions and flow rates along pipes are constrained by topological features such as connectivity, flow conservation, and loop structures Kerimov et al. (2025); Money et al. (2023). Similar dependencies appear in brain networks, where neural activity propagates along anatomical pathways Bispo et al. (2025), and in epidemiological models, where disease transmission follows contact networks Fan et al. (2022). Capturing this higher-order structural context, alongside autoregressive and non-stationary temporal dynamics, is essential for accurate modeling. However, most existing approaches address these aspects separately and lack a unified, scalable framework. In this work, we propose a comprehensive approach to multivariate time-series modeling that jointly incorporates four key components: (i) nonlinear dependencies, (ii) higher-order structural information, (iii) autoregressive temporal dynamics, and (iv) online adaptability to non-stationary environments. To effectively capture higher-order structural relationships that go beyond simple pairwise interactions, we build on the framework of Abstract Cellular Complexes (ACCs) Wasserman (2018). ACCs extend traditional graph-based representations by allowing data to be associated not only with nodes and edges, but also with higher-dimensional cells such as faces and volumes. This enables a more expressive and topologically grounded representation of complex relational structures present in many real-world systems. To capture nonlinear dependencies within this framework, we adopt an additive kernel-based autoregressive model that balances expressiveness with interpretability. Furthermore, to accommodate dynamic environments and streaming data, we propose an efficient online learning algorithm equipped with formal performance guarantees in terms of dynamic regret.

### 1.1 RELATED WORK

**Graph based models** Graph-based approaches typically assume pairwise relationships between time-series and often overlook the higher-order topological structures present in many real-world networks Ortega et al. (2018); Sandryhaila & Moura (2014). While graph neural networks (GNNs)

have become a popular approach for time-series modeling Wu et al. (2020); Cheng et al. (2022), especially for batch data, linear models and kernel methods have shown promising results in the context of online learning Natali et al. (2020); Zaman et al. (2021); R. Money et al. (2023). In Natali et al. (2020); Zaman et al. (2021), linear relationships between nodes are assumed, with models being learned online. In contrast, R. Money et al. (2023) presents an online nonlinear model based on kernels to capture dependencies. However, these works do not account for the higher-order topological relationships present in many real-world systems.

**Higher order models**    Hypergraphs, simplicial complexes (SCs), and cell complexes (CCs) serve as common topological frameworks for modeling and analyzing complex multi-way relationships. While hypergraphs offer significant flexibility, their algebraic representations are often intractable, limiting their practical applicability Isufi et al. (2025). SC- and CC-based approaches provide more structured representations and leverage Hodge Laplacians to encode adjacency relationships between entities at the same and different levels across dimensions Isufi et al. (2025). For example, Krishnan et al. (2024) employs Hodge Laplacian-based simplicial convolutions to capture spatial dependencies in SCs, which are restricted to simplex-based constructions with strict attachment rules. In contrast, Marinucci et al. (2024) introduces adaptive filters over cell complexes, a generalization of SCs that supports more diverse cell types and flexible attachment rules, enabling richer representations of complex topologies in time-series modeling. However, Hodge Laplacian-based linear models often struggle to capture the nonlinear relationships commonly observed in real-world data. To address this, topology-aware neural architectures have been proposed, incorporating mechanisms such as message passing Bodnar et al. (2021b), convolution Yang et al. (2022); Huang et al. (2024), and attention Hajij et al. (2022); Battiloro et al. (2024). While these methods are expressive and powerful, their application to time-series data remains relatively underexplored. Also, it is not trivial to train these neural networks for very long multivariate time-series that may not be stationary in terms of the existing interdependencies. Moreover, they are typically designed for offline training and may lack robustness in dynamic or streaming settings, where tracking guarantees and adaptability are critical. Overcoming these limitations is essential for developing practical and efficient models for structured time-series modeling. This work addresses the problem of modeling multivariate time-series defined over higher-order topological structures in dynamic, streaming environments.

Our main contributions are listed as follows:

**Structure-aware modeling with Abstract Cellular Complexes (ACCs)**    We propose a framework that leverages ACCs to represent signals defined over nodes, edges, and higher-dimensional cells. This enables the modeling of complex topological relationships that go beyond pairwise interactions, capturing richer dependencies in multivariate time-series data.

**Scalable autoregressive model in RKHS over higher-order networks**    We introduce an additive, kernel-based autoregressive formulation that captures nonlinear dependencies among higher-order entities. Our approach not only captures complex multiway dependencies but also enables the decomposition of the overall model into parallelizable submodules, facilitating efficient training and scalability.

**Online learning with theoretical performance guarantees**    We develop an online optimization algorithm for learning the proposed model in streaming settings. We provide theoretical guarantees in terms of dynamic regret, ensuring robust adaptation to non-stationary environments.

To the best of our knowledge, this is the **first work on online learning of nonlinear processes over the cellular complexes.**

**Definitions and Notation**    In this section, we will define the notions in an abstract way and introduce the notation. Advanced treatment of the concepts are given in Appendix.

**Definition 1** (Abstract Cell Complex). *An* abstract cell complex (ACC)*, denoted by $\mathbb{W}$, is a way of representing a space in terms of basic building blocks, called cells, that are organized by dimension and by how they connect to each other. Cells of dimension $0$ correspond to nodes, cells of dimension $1$ to edges, cells of dimension $2$ to polygons, and so on. The boundary relation specifies how lower-dimensional cells are connected to higher-dimensional ones, and the dimension function simply assigns the level (or dimension) of each cell. Collectively, these ingredients allow us to de-*

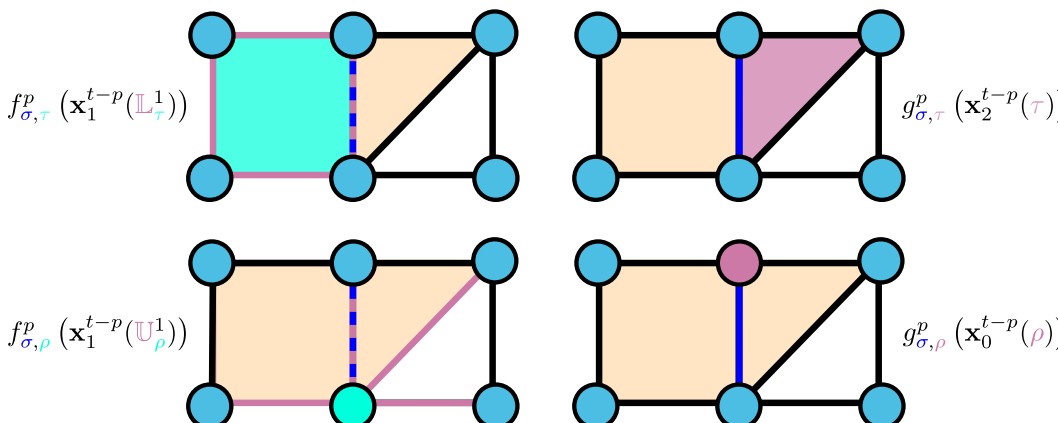

Figure 1: An illustration of upper and lower adjacent elements of NL-HORSO model for each of the four elements, namely, (from left to right and up to down) upper self-transformation $f^p_{\sigma,\tau}\left(\mathbf{x}^{t-p}_k(\mathbb{L}^k_\tau)\right)$, upper transformation $g^p_{\sigma,\tau}\left(\mathbf{x}^{t-p}_{k+1}(\tau)\right)$, lower self-transformation $f^p_{\sigma,\rho}\left(\mathbf{x}^{t-p}_k(\mathbb{U}^k_\rho)\right)$ and lower transformation $g^p_{\sigma,\rho}\left(\mathbf{x}^{t-p}_{k-1}(\rho)\right)$. The elements are color coded to show different properties. Blue is the element in question, pink ones pass their information to blue, and the cyan elements connect blue and pink elements. Dotted line means the element is both taking in and passes the information to itself. A more general diagram considering time lags is available in Appendix.

*scribe complex structures, such as graphs, meshes, or higher-dimensional objects, in a unified and systematic way.*

An example of a 2-dimensional ACC can be seen in Figure 1, where there are 2 polygons, 8 edges and 6 nodes that constitutes an ACC. We can also define the sets showing the connections of the ACC using its topology. This will be useful in the definition of the method.

**Definition 2** (Upper and Lower Level Sets)**.** *Given a cell $\sigma$ and a dimension $k$, the $k$-lower level set of $\sigma$, $\mathbb{L}^k_\sigma$, is the collection of all $k$-dimensional cells that sit just below $\sigma$ (its $k$-faces), while the $k$-upper level set of $\sigma$, $\mathbb{U}^k_\sigma$, is the collection of all $k$-dimensional cells that sit just above $\sigma$ (its $k$-cofaces). In simple terms, these sets describe the neighbors of $\sigma$ that are either one dimension smaller or one dimension larger.*

For each set of $k$-cells, we can attach values called *$k$-cellular signals*. Collecting the values across all $k$-cells gives a signal vector $\mathbf{x}_k$. If we only care about some subset of cells $\mathbb{S}$, we just denote the aggregation of the values residing over that subset as $\mathbf{x}_k(\mathbb{S})$. When the signal changes over time, we record its values at different time steps in a signal matrix $\mathbf{X}_k$, where each column corresponds to one time instant, and at each time instant the corresponding value is shown as $\mathbf{x}^t_k$.

**Definition 3** (Connection Reproducing Kernel Hilbert Spaces)**.** *The connection reproducing kernel Hilbert space (RKHS) is defined on the connections between adjacent cells $(\sigma, \tau)$, where $\tau$ is either a face or a coface of $\sigma$. For each such pair and for each natural number $p$, we associate two kernel spaces, denoted by $\mathbb{F}^p_{\sigma,\tau}$ and $\mathbb{G}^p_{\sigma,\tau}$, which capture how functions supported on $\sigma$ and on $\tau$ interact through their adjacency. Intuitively, $\mathbb{F}^p_{\sigma,\tau}$ governs functions on $\sigma$ as viewed from $\tau$, while $\mathbb{G}^p_{\sigma,\tau}$ governs functions on $\tau$ as viewed from $\sigma$, and together they encode the flow of information across the connection. The overall connection RKHS is then the collection of all such spaces over all adjacent cell pairs.*

The role of $p$ when defining the connection RKHS is the account for the number of features defined over the whole structure. The main idea in RKHS is to learn the parameters that construct a function within the RKHS. Since the number of parameters scale as the number of time instants, we will use the following notation: for a learnable parameter or function over $\theta$ representing the relationships between the pair $(\sigma, \tau)$ of an ACC, we denote the aggregation of parameters related to that relationship as $\theta_{\sigma,\tau}$. Moreover, the aggregated learnable parameter over cell pairs having the same $k$-dimensional cell $\sigma$ is denoted as $\theta_\sigma$.

## 2 NONLINEAR PROCESSES OVER ABSTRACT CELLULAR COMPLEXES

Abstract Cellular Complexes (ACCs) provide a mathematically tractable framework for representing the higher-order topological structure of data-generating processes. Unlike traditional graph-based models that define signals over nodes, ACCs support signal definitions over nodes, edges, and higher-dimensional cells Isufi et al. (2025); Schaub et al. (2021). These multi-scale signals are interrelated through well-defined boundary maps, making ACCs a powerful inductive bias for structured time-series modeling with fewer parameters. However, most existing work on ACCs assumes linear dynamics and lacks adaptability to streaming data. While neural models over ACCs exist, they are often computationally expensive, offer limited interpretability or theoretical guarantees and not suited to online learning. Motivated by these limitations, we propose a lightweight, nonlinear autoregressive model over ACCs, formulated in a reproducing kernel Hilbert space (RKHS), and which we introduce formally next.

Consider the finite ACC $\mathbb{W} = (\mathbb{X}, \dim, \prec_b)$, and a random vector process $\mathbf{x}^t$ over the cells of $\mathbb{W}$, $\mathbf{x}_k^t$ denotes the random values belonging to the $k$-dimensional cells. We consider a $P$-th order autoregressive functional model of the form

$$
\begin{aligned}
\mathbf{x}_k^t(\sigma) = \sum_{p=1}^{P} \Bigg[ & \sum_{\tau \in \mathbb{U}_\sigma^{k+1}} f_{\sigma,\tau}^p \left( \mathbf{x}_k^{t-p}(\mathbb{L}_\tau^k) \right) + g_{\sigma,\tau}^p \left( \mathbf{x}_{k+1}^{t-p}(\tau) \right) \\
& + \sum_{\rho \in \mathbb{L}_\sigma^{k-1}} f_{\sigma,\rho}^p \left( \mathbf{x}_k^{t-p}(\mathbb{U}_\rho^k) \right) + g_{\sigma,\rho}^p \left( \mathbf{x}_{k-1}^{t-p}(\rho) \right) \Bigg] + \mathbf{w}_k^t(\sigma)
\end{aligned}
\tag{1}
$$

for a given $k \in \{0, 1, \ldots, K\}$ and $\sigma \in \mathbb{W}_k$. Here, the function called upper/lower self-transformation $f_{\sigma,c}^p(.)$ captures the relationship between the time-series value at cell $\sigma$ and the $p$-th time-lagged values of cells that are horizontally adjacent through cell $c$. Similarly, upper/lower transformation $g_{\sigma,c}^p(.)$ encodes the relationship between the $t$-th time-series value at cell $\sigma$ and the $p$-th time-lagged value of a vertically adjacent cell $c$ and $\mathbf{w}_k^t(\sigma)$ represents the noise in modeling the time-series over cell. Note that here, depending on the arguments of $f_{\sigma,c}^p(.)$ and $g_{\sigma,c}^p(.)$, we can represent both upper and lower adjacent relationships. For example, an edge signal can be expressed as a nonlinear combination of adjacent edge signals (connected via shared nodes or polygons), as well as adjacent node and polygon signals (see Figure 1).

Notice that; first, in equation 1, we model the multivariate time series in a cell-separable manner, where the $t$-th time-series value of each cell depends only on its immediate neighbors. This structure facilitates the decomposition of a high-dimensional problem into smaller, more tractable subproblems, thereby significantly reducing computational complexity. This is achieved by systematically aggregating information from neighboring cells, effectively leveraging the higher-order topological structure inherent in the ACC representation. Second, in equation 1, we use the topological information coming from the immediate dimensions, i.e. for $k$-dimensional cells we use $k+1$, and $k-1$ dimensional information. This is different from the works in topological machine learning Giusti et al. (2023); Bodnar et al. (2021a); Hajij et al. (2020), where the authors extract cellular complexes from simpler graph structures to make use of their richer representation, which is called lifting. In this work, we assume no lifting, rather **the data is already defined over the cellular complex**.

Now we assume that the nonlinear functions $f_{\sigma,c}^p(.)$ and $g_{\sigma,c}^p(.)$ belong to the connection RKHS $\mathbb{F}_{\sigma,c}^p$ and $\mathbb{G}_{\sigma,c}^p$, respectively. Our aim is to estimate the functions $f_{\sigma,c}^p(.)$ and $g_{\sigma,c}^p(.)$. As the functions belong to RKHS, the functions can be expressed as an infinite sum of kernel expansions. This allows us to formulate a nonparametric optimization problem to estimate the unknown functions. The Representer Theorem guarantees that the solution to this nonparametric problem can be expressed as a finite combination of kernel evaluations, where the number of kernel evaluations is equal to the number of data samples Schölkopf & Smola (2002). With the help of the Representer Theorem, without loss of generality for each function centered at some adjacent cell tuple $(\sigma, \tau)$ we have the following representation with a finite number of terms

$$
f_{\sigma,\tau}^p (\boldsymbol{x}_k) = \sum_{t=p}^{T-p+1} \beta_{\sigma,\tau}^{p,t} \kappa_{\sigma,\tau}^p \left( \boldsymbol{x}_k, \boldsymbol{x}_k^{t-p}(\mathbb{L}_\tau^k) \right), \quad g_{\sigma,\tau}^p (\boldsymbol{x}_{k+1}) = \sum_{t=p}^{T-p+1} \delta_{\sigma,\tau}^{p,t} \lambda_{\sigma,\tau}^p \left( \boldsymbol{x}_{k+1}, \boldsymbol{x}_{k+1}^{t-p}(\tau) \right) \tag{2}
$$

where $\beta_{\sigma,\tau}^{p,t}$ and $\delta_{\sigma,\tau}^{p,t}$ are coefficients of the kernel function centered at adjacent cell tuple $(\sigma, \tau)$ at time $t$, and $\kappa_{\sigma,c}^p(.,.), \lambda_{\sigma,c}^p(.,.)$ are the kernels inducing the RKHS $\mathbb{F}_{\sigma,c}^p, \mathbb{G}_{\sigma,c}^p$, and $T$ is the total number of data samples. A similar expression for the lower adjacency can be written.

## 3 ONLINE LEARNING OF NONLINEAR KERNEL MODELS

In this section, an online learning framework for estimating the kernel coefficients introduced in equation 2 is considered. We propose a unified framework for solving online kernel learning problems over cellular complexes, enabling usage of the algorithm over the range of different assumptions over the model in equation 1 and kernel representation in equation 2.

Let $\mathbf{x}_k^t[\sigma] = F_k^P \left( \{\mathbf{x}^s\}_{s=t-P}^{t-1}, \theta_\sigma^t \right) [\sigma]$ be a shorthand representation of equation 1 for all $\sigma \in \mathbb{W}_k$, with the kernel representation in equation 2 introduced, operating over $k$-dimensional cells, where $P$ is the maximum time lag of the model and $\theta_\sigma^t$ is the aggregation of variables $\beta_{\sigma,\tau}^{p,t}, \beta_{\sigma,\rho}^{p,t}, \delta_{\sigma,\tau}^{p,t}, \delta_{\sigma,\rho}^{p,t}$. Here we observe $\{\mathbf{x}^s\}_{s=0}^t$ in a streaming fashion, we aim to find an online estimate of time varying parameter $\theta_\sigma^t \ \forall \ \sigma$ at time $t$. Notice that in general, we don't assume any stationarity in the set of kernel coefficients denoted by $\theta_\sigma^t$. In this framework, the problem of modeling a multivariate time-series $\mathbf{x}_k^t \in \mathbb{R}^{N_k}$ is decomposed into $N_k$ subproblems, as the parameter $\theta_\sigma^t$ associated with a particular cell $\sigma$ is independent of the parameters corresponding to other time series. This decomposition enables parallelization of the optimization process. In the remainder of this section, we formulate the solution for estimating the model parameters for a specific cell $\sigma$; however, the approach is extendable to entire multivariate time series.

To find the optimal parameters $\theta_\sigma^t{}^*$ that govern the dynamics of time-series $\boldsymbol{x}_k^t(\sigma)$ in an online fashion, we formulate a regularized recursive least squares optimization problem. This formulation allows the model to be updated whenever a new data sample becomes available by solving the optimization problem :

$$
\theta_\sigma^t{}^* = \arg\min_{\theta_\sigma} \sum_{\sigma \in \mathbb{W}_k} h_\sigma^t (\theta_\sigma, \kappa_\sigma)
$$

$$
\text{where } h_\sigma^t (\theta_\sigma, \kappa_\sigma) := \mu \sum_{s=P}^t \gamma^{t-s} \left\| \mathbf{x}_k^t - F_k^P \left( \{\mathbf{x}^s\}_{s=t-p}^{t-1}, \theta_\sigma \right) \right\|_2^2 + \lambda w(\theta_\sigma)
$$

(3)

where $\kappa_\sigma$ is the aggregated representation features generated by kernels. The objective function consists of a data fitting term and a non-decreasing regularizer $w(\theta_\sigma)$. Note that the regularizer is typically non-differentiable in order to enforce sparsity. The parameter $\gamma$ is the forgetting factor, which controls the exponentially decaying influence of the error from previous time instances. The parameter, $\mu = 1 - \gamma$ can be seen as representing the effective weight given to newly observed data, allowing the parameters to adapt to a changing environment while prioritizing recent observations.

In our model construction, we consider a regularization function $w(\theta_\sigma)$ that controls both the sparsity of the parameters and their conformity to the underlying cellular complex $\mathcal{W}$. To achieve this, we decompose the regularization function $w(\theta_\sigma)$ into two components, corresponding to the coefficients associated with upper and lower adjacent cells. Furthermore, to enforce a unified topological representation, we introduce conformity constraints on the coefficients related to both self-transformations and inter-cell transformations.

We begin by grouping the parameters corresponding to functions that capture the influence of a particular cell $\tau$ on cell $\sigma$. We denote this group of parameters with vector $\theta_{\sigma,\tau}^{p,t}$, which aggregates coefficients that characterize the functions $f_{\sigma,\tau}^p(.)$ and $g_{\sigma,\tau}^p(.)$. Based on this grouping, we define a Group Lasso regularizer to promote structured sparsity and topological coherence in the parameter space:

$$
w_1(\theta_\sigma) = \sum_{p=1}^P \sum_{\tau \in \mathbb{U}_\sigma^{k+1}} \|\theta_{\sigma,\tau}^p\|_2, \quad w_{-1}(\theta_\sigma) = \sum_{p=1}^P \sum_{\rho \in \mathbb{L}_\sigma^{k-1}} \|\theta_{\sigma,\rho}^p\|_2.
$$

(4)

The functions $w_1(.)$ and $w_{-1}(.)$ represent the regularization terms corresponding to upper and lower adjacencies, respectively. The regularizer $w_1(.)$ encourages the inclusion of coefficients only when

there is strong evidence that for time-series modeling of a cell $\sigma$ its upper adjacent cell $\tau$ should be present within the ACC structure; otherwise, the coefficients are suppressed by the Group Lasso penalty. Similarly, $w_{-1}(.)$ promotes sparsity by ensuring that only the most relevant lower adjacencies contribute to the model. Now, equation 3 can be rewritten as

$$\theta_\sigma^{t\,*} = \underset{\theta_\sigma}{\arg\min}\ l_\sigma^t(\theta_\sigma) + \lambda_1 w_1(\theta_\sigma) + \lambda_{-1} w_{-1}(\theta_\sigma) \tag{5}$$

where $l^t(.) = \mu \sum_{s=P}^t \gamma^{t-s} \left\| \mathbf{x}_k^t - F_k^P(.,.) \right\|_2^2$ is the data fidelity term in equation 3. To solve the optimization problem, we need to minimize a convex loss function and a non-differentiable regularizer. Composite objective mirror descent (COMID) Duchi et al. (2010) provides an efficient solution for such an optimization problem. The online COMID update for the optimization problem corresponding to cell $\sigma$ can be given as:

$$\check{\theta}_\sigma^{t+1} = \arg\min_{\theta_\sigma} J_\sigma^t(\theta_\sigma), \tag{6}$$

where

$$J_\sigma^t(\theta_\sigma) \nabla \ell_\sigma^t(\tilde{\theta}_\sigma^t)^\top (\theta_\sigma - \tilde{\theta}_\sigma^t) + \frac{1}{2\gamma_t} \|\theta_\sigma - \tilde{\theta}_\sigma^t\|_2^2$$
$$+ \sum_{p=1}^P \left[ \lambda_u \sum_{\tau \in \mathbb{U}_\sigma^{k+1}} \|\theta_{\sigma,\tau}^p\|_2 + \lambda_l \sum_{\rho \in \mathbb{L}_\sigma^{k-1}} \|\theta_{\sigma,\rho}^p\|_2 \right]. \tag{7}$$

The objective function $J_\sigma^t(.)$ consists of three components. The first term guides the solution in the direction of the gradient of the loss function, $\nabla \ell_\sigma^t(\tilde{\theta}_\sigma^t)$, promoting alignment with the current data. The second term is a Bregman divergence Gutmann & Hirayama (2011), which provides stability to the online updates by encouraging the solution to remain close to the previous estimate. Finally, the third term is a regularization component that promotes sparsity in the solution. In equation 7, $\tilde{\theta}_\sigma^t$ is the previous coefficient set either by itself or padded with zeros, and $\gamma_t$ is the step size. A closed-form solution for the COMID update can be obtained via multidimensional soft thresholding as,

$$\check{\theta}_{\sigma,c}^{p,t+1} = \left( \tilde{\theta}_{\sigma,c}^{p,t} - a_t \boldsymbol{q}_{\sigma,c}^{p,t} \right) \cdot \left[ 1 - \frac{a^t \lambda_c}{\|\tilde{\theta}_{\sigma,c}^{p,t} - a^t \boldsymbol{q}_{\sigma,c}^{p,t}\|_2} \right]_+ \tag{8}$$

Here, $\mathbf{q}_{\sigma,c}^{p,t} = \nabla \ell_\sigma^t(\tilde{\theta}_{\sigma,c}^{p,t})$ denotes the component of the gradient of the loss function corresponding to the parameters $\tilde{\theta}_{\sigma,c}^{p,t}$. For all $c \in \mathbb{U}_\sigma^{k+1} \cup \mathbb{L}_\sigma^{k-1}$ and $p \in \{1, 2, \ldots, P\}$, the regularization weight $\lambda_c$ is defined as $\lambda_1$ if $c \in \dim^{-1}(\{k+1\})$, and $\lambda_{-1}$ otherwise. The positive part operator is defined as $[\cdot]_+ = \max\{0, \cdot\}$.

Equation 8 describes the parameter update rule for modeling the nonlinear influence of a neighboring cell $c$ on the target cell $\sigma$ at time $t$. This update step can be applied across all relevant neighboring and target cells to construct the full model for the multivariate time series. We refer to this overall approach for nonlinear modeling of multivariate time-series as *Non-linear Higher-Order Recursive Sparse Online Learning* (NL-HORSO). The steps of the NL-HORSO algorithm are outlined in Algorithm 1.

Although NL-HORSO provides effective solutions, its performance can degrade as the length of the time series increases, since the dimensionality of the kernel space grows with the time horizon. Several standard techniques have been proposed in the literature to address this issue. We present one such approach in the Appendix, where we leverage Random Fourier Features Rahimi & Recht (2007a); Shen et al. (2019) to obtain a fixed-dimensional approximation of the kernel. We refer to this variant as *Random Feature-based Higher-Order Recursive Sparse Online Learning* (RF-HORSO). In the next section, we demonstrate the tracking capability of the proposed algorithm RF-HORSO by establishing a dynamic regret.

## 4 DYNAMIC REGRET ANALYSIS OF RF-HORSO

For a time series defined over cell with dimension $k$, our theoretical analysis of dynamic regret is based on the assumptions : (A1) time series samples have bounded norm with parameter $B_x$, (A2)

| $\nu$ | 0.01 | 0.1 | 1 | 10 |
|---|---|---|---|---|
| $p_1$ | $0.077 \pm 0.002$ | $0.074 \pm 0.002$ | $0.076 \pm 0.003$ | $0.084 \pm 0.004$ |
| $p_2$ | $0.005 \pm 0.002$ | $0.005 \pm 0.002$ | $0.005 \pm 0.002$ | $0.005 \pm 0.002$ |
| $p_3$ | $0.250 \pm 0.007$ | $0.246 \pm 0.007$ | $0.238 \pm 0.008$ | $0.248 \pm 0.006$ |
| $p_4$ | $0.005 \pm 0.002$ | $0.005 \pm 0.002$ | $0.005 \pm 0.002$ | $0.005 \pm 0.002$ |
| $D$ | 1 | 2 | 4 | 9 |
| $p_1$ | $0.078 \pm 0.003$ | $0.077 \pm 0.003$ | $0.076 \pm 0.004$ | $0.076 \pm 0.004$ |
| $p_2$ | $0.005 \pm 0.002$ | $0.005 \pm 0.002$ | $0.005 \pm 0.002$ | $0.005 \pm 0.002$ |
| $p_3$ | $0.240 \pm 0.008$ | $0.239 \pm 0.008$ | $0.238 \pm 0.008$ | $0.237 \pm 0.008$ |
| $p_4$ | $0.005 \pm 0.002$ | $0.005 \pm 0.002$ | $0.005 \pm 0.002$ | $0.005 \pm 0.002$ |

(a) Synthetic data ablation results for kernel bandwidth and number of features

| $\nu_p$ | AUC |
|---|---|
| 0.01 | $0.9265 \pm 0.0643$ |
| 0.1 | $0.9485 \pm 0.0495$ |
| 0.5 | $0.9524 \pm 0.0378$ |
| 1.0 | $0.9585 \pm 0.0403$ |
| 2.0 | $0.9328 \pm 0.0422$ |
| 3.5 | $0.9185 \pm 0.0359$ |
| 5.0 | $0.9012 \pm 0.0291$ |

(b) AUC values for different Gaussian bandwidths

Table 1: Synthetic Data Ablation and AUC results

kernels are shift-invariant, (A3) basis generating the gradient $\boldsymbol{q}_c$ contains at least $\rho_l$ power in its elements, (A4) that the power of basis elements generating the gradient is bounded by $L$, (A5) the cost function $h_\sigma^t$ has Lipschitz continuity parameter $L_h$, (A6) $a^t = 1/L$ (See appendix for more details and significance of each assumptions).

**Dynamic Regret:** Dynamic regret provides a quantitative measure of the tracking capability of an online algorithm, it is defined as the cumulative sum of the difference between the estimated cost function and the optimal cost function. At time $T$, for each $\sigma \in \mathbb{W}_k$, it can be expressed as:

$$R_\sigma^{RF}[T] = \sum_{t=P}^{T-1} \left[ h_\sigma^t(\theta_\sigma^t, \kappa^t) - h_\sigma^t(\hat{\theta}_\sigma^t, \hat{\kappa}^t) \right]. \tag{9}$$

Here, $\hat{\kappa}^t$ and $\hat{\theta}_\sigma^t$ denote the collection of random features at time $t$ and the corresponding weights estimated by COMID in the approximated feature space. In contrast, $\kappa^t$ and $\theta_\sigma^t$ refer to the features generated by the original kernel evaluations and the corresponding optimal parameters. We explicitly distinguish the original kernel evaluations because the optimization is performed in a different space, the random feature space, rather than in the original reproducing kernel Hilbert space (RKHS). In RF-HORSO, we first approximate the infinite-dimensional RKHS using a fixed-dimensional random feature representation, and then apply the proposed online learning algorithm. As a result, the dynamic regret expression in equation 9 accounts for both the approximation error due to random features and the tracking error from online optimization.

From equation 9, it is evident that the growth rate of dynamic regret serves as a direct indicator of the algorithm's tracking capability. If the dynamic regret grows sublinearly, the estimated solution asymptotically converges to the optimal one, demonstrating strong adaptability to non-stationary environments. A linear growth in regret suggests convergence to a steady-state solution with a bounded error relative to the optimum. On the other hand, superlinear growth in dynamic regret indicates poor tracking performance, as errors accumulate over time and the model struggles to adapt to changes in the underlying data distribution.

We also define $\hat{\theta}_\sigma^{t*}$ as the minimizer of the cost function $h_\sigma^t$ under the random feature approximation. Using this, we define the path length of the optimal sequence in the random feature space as:

$$W_\sigma[T] = \sum_{t=P}^{T-1} \left\| \hat{\theta}_\sigma^{t*} - \hat{\theta}_\sigma^{t-1*} \right\|_2. \tag{10}$$

This quantity captures the cumulative variation of the optimal parameters over time and serves as a measure of non-stationarity in the target sequence. A smaller path length indicates a more stable or slowly changing environment, whereas a larger path length reflects rapid temporal variation, which poses a greater challenge for online learning algorithms to track effectively.

Next, we provide the dynamic regret for the RF-HORSO algorithm in Theorem 1.

**Theorem 1.** *Given the assumptions A1-A6, $\exists \epsilon, C > 0$ such that RF-HORSO satisfies the following bound for the regret:*

$$R_\sigma^{RF}[T] \in \mathcal{O}\left( \frac{L\sqrt{B_x}}{\rho_l} W_\sigma(T) + \sqrt{T} L_h \right) \tag{11}$$

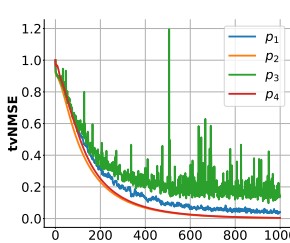 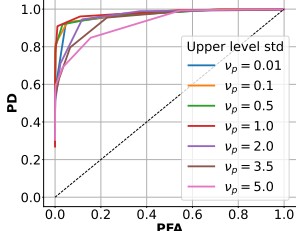 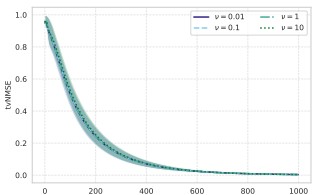

(a) tvNMSE plot with respect to different types of nonlinearities

(b) ROC curve for different upper level standard deviation $\lambda_1$

(c) tvNMSE plot with respect to different kernel bandwidths for $p_4$

Figure 2: Synthetic dataset results.

Please refer to Appendix for the proof sketch and additional terms omitted here. Dynamic regret here consists of two parts. The first part, which comes from the difference between optimal random feature parameters and the learned parameters, depends on the path length of the optimal parameters, the power distribution of the gradient updates, and finally the bound on the time-series instances. Here the bound increases as the path length increases, i.e., the regret is larger when the environment is highly non-stationary, it increases as the power distribution of the gradient updates are uneven which signals non-smooth environments and finally, it relates to the bound on the signal power, where regret is expected to scale with it. The increase of error/regret is also shown with real/synthetic experiments where in Figure 2a, it is shown that the regret becomes more unstable as the underlying function becomes non-smooth and in Figure 3c, it is shown that the algorithm falls behind as the medium has seasonal changes where it learns if the environment is stationary for some time on a real dataset. In the second part, the regret bound increases sublinearly which is a desired regret bound for online scenarios, and it is scaled by the Lipschitz continuity of the objective function, where a quadratic cost with Group Lasso regularizer utilized throughout the work.

## 5 EXPERIMENTAL RESULTS

We evaluate the capabilities of RF-HORSO using three types of cellular complex datasets. First, we demonstrate the sensitivity of the algorithm with respect to estimation performance on a synthetic cellular complex with varying nonlinear functions. Next, we assess our algorithm using Navier-Stokes data Baratta et al. (2023) to showcase RF-HORSO's competence in modeling real-world nonlinear processes. Detailed settings and information for each dataset are provided in the Appendix. Finally, we evaluate its performance on the Ocean Dataset NOAA CoastWatch (2024a); Copernicus Marine Service (2024); NOAA CoastWatch (2024b). For benchmarking, we consider "online" graph and cellular complex methods such as TIRSO Zaman et al. (2021), TopoLMS Marinucci et al. (2024) , S-VAR Krishnan et al. (2023), and SC-VAR Krishnan et al. (2024). We refer interested readers to Appendix for the selection criteria of the datasets and the baseline methods.

**Synthetic Data Experiments** We evaluate the sensitivity of RF-HORSO to various hyperparameters, including the number of random features $D$, Gaussian kernel standard deviation $\nu$, regularization strength, percentage of holes, and different nonlinearities. The nonlinearities considered are:

$$p_1(x) = 0.25\cos(x^2) + 0.25\cos(2x) + 0.5\cos(x), \quad p_2(x) = \frac{1}{\sqrt{2\pi}}e^{-x^2/2}$$

$$p_3(x) = 0.25\sin\left(\sqrt{|x|}\right) + 0.25\sin(2x) + 0.5\sin(x), \quad p_4(x) = 0.25\sin(0.2x) + 0.5\sin(0.1x)$$

$$p_5(x) = \frac{1}{\sqrt{2\pi\nu_p}}e^{-x^2/2\nu_p}$$

Details on their application to cellular complexes are provided in the Appendix. As shown in Table 1 and Figures 2a, 2c, RF-HORSO is robust to changes in $D$, the kernel bandwidth $\nu$, and the presence of holes due to its expressive power via upper and lower adjacencies. However, performance is sensitive to the choice of nonlinearity, as predicted by Theorem 1. When $\nu$ fails to capture the signal's frequency content, tracking degrades. Higher-frequency nonlinearities (e.g., $p_1$, $p_3$) induce greater tvNMSE fluctuations, whereas smoother functions like $p_2$ yield more stable behavior,

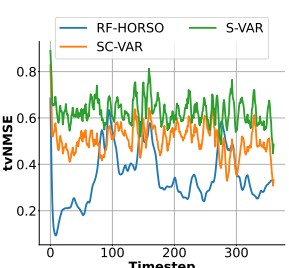
(a) Comparison of different on-line algorithms over Ocean Cellular Dataset

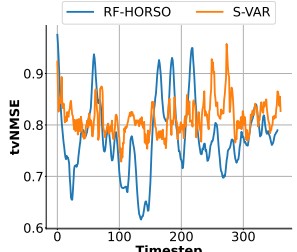
(b) Comparison of different on-line algorithms over Ocean Edge Dataset

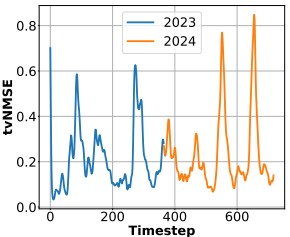
(c) tvNMSE of RF-HORSO on Ocean Cellular Dataset for 2023-2024 period

Figure 3: Real dataset results.

consistent with Bochner's theorem, which connects kernel smoothness with the spectral properties of random features Rahimi & Recht (2007b).

In addition, we compute the AUC-ROC for different upper-level Gaussian bandwidths $\nu_p$ by thresholding the learned weights in Table 1. The results indicate that RF-HORSO is able to recover the underlying topological structure, suggesting that the algorithm is not only fitting the signal but also becoming aware of the topology governing the data.

**Cellular Complex Based Datasets**  We next evaluate RF-HORSO on real-world datasets where the underlying dynamics are naturally supported on cellular complexes. Physical processes such as fluid dynamics and ocean circulation evolve on manifolds, making cellular complexes a suitable representation. Detailed dataset descriptions are provided in the Appendix.

Figure 3 summarizes the results. On the **Navier–Stokes dataset**, RF-HORSO reliably tracks both horizontal and vertical velocity components, confirming its ability to adapt to nonlinear and non-stationary physical processes.

On the more challenging **Ocean Cellular Dataset**, RF-HORSO achieves stronger accuracy than online baselines such as S-VAR and SC-VAR (Figure 3a). Notably, the tvNMSE curve exhibits peaks aligned with seasonal transitions, suggesting that the model captures recurring large-scale shifts in ocean dynamics. To further validate this, we trained RF-HORSO on a two-year sequence (2023–2024). The results (Figure 3c) show four clear error peaks in 2024, consistent with seasonal cycles, whereas the 2023 sequence shows three pronounced peaks, with the fourth attenuated due to training starting mid-season. These findings indicate that RF-HORSO can uncover structured temporal patterns that linear baselines fail to capture. Finally, on the **Ocean Edge Dataset** (Figure 3b), RF-HORSO and S-VAR perform comparably, as SC-VAR is identical to S-VAR in the absence of node and triangular signals. Here, RF-HORSO maintains a slight advantage, demonstrating robustness across both edge- and cell-level signal representations.

For completeness, we also evaluated TIRSO and TopoLMS under the same experimental setup. In all cases, their learned parameter matrices collapsed to the identity, effectively reproducing past values without learning new dynamics. This outcome is consistent with their linear assumptions, which limit adaptability in nonlinear, non-stationary settings. For clarity of presentation, we omit their curves from the figures but report quantitative results in the Appendix.

## 6 CONCLUSION

In this work, we introduced an online learning framework for streaming multivariate time series defined over cellular complexes using kernel methods. We first proposed NL-HORSO, a general algorithm for learning kernel coefficients directly from cellular complex signals. To improve scalability, we formulated a randomized approximation, RF-HORSO, based on the random features assumption. We then analyzed the dynamic regret of RF-HORSO in the online setting, providing theoretical guarantees. Experimental results indicate that the interplay between the estimator, the data structure, and the topological representation has a significant impact on performance, highlighting the need for further study into adaptive and data-driven topological modeling.

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

## A    Detailed Definitions and Explanations

### A.1    Abstract Cellular Complexes

In this work, for the sake of clarity, we consider signals supported on a 2-dimensional cell complex, which consists of three fundamental components: nodes, edges, and polygons [1]. Nodes represent the fundamental elements, while edges are defined as two-tuples of nodes, forming connections between them. Polygons are higher-order structures consisting of three or more nodes that are interconnected via edges. To formalize these structures, we employ the framework of ACCs, which provides a general and flexible foundation encompassing graphs, simplicial complexes, and other geometric structures Wasserman (2018).

**Definition 4** (Abstract Cell Complex (Sardellitti et al., 2021; Klette, 2000)). *An ACC is a tuple* $\mathbb{W} = \{\mathbb{X}, \prec_b, dim\}$, *which is composed of the set* $\mathbb{X}$ *together with a strict partial order* $\prec_b$ *called boundary relation and a non-negative dimension function dim* : $\mathbb{X} \to \mathbb{Z}$, *which is monotone with respect to this boundary relation and the usual order on* $\mathbb{Z}$.

A $k$-cell of $\mathbb{W}$, is an element $x \in \mathbb{X}$ with $\dim(x) = k$, where $0-$cells are termed as nodes, $1-$cells as edges and $2-$cells as polygons. The set of all $k$-cells is denoted as $\mathbb{W}_k$, and the number of $k$-cells is denoted by $N_k$. We call $K$ as the dimension of the cellular complex $\mathbb{W}$ if there exists $K = \max_{\sigma \in \mathbb{W}} \dim(\sigma)$. Throughout this text, we assume that $K$ exists and is finite. An example of an ACC with $K = 2$ is illustrated in Figure 1. For a given ACC $\mathbb{W} = (\mathbb{X}, \dim, \prec_b)$, we can define lower and upper level sets which corresponds to the set of cells that are adjacent to some set on $k$-dimensions, with the help of the binary relation $\prec_b$ defined as follows:

**Definition 5** (Upper and Lower Level Sets). *Given a cell* $\sigma \in \mathbb{X}$ *and an integer* $k \in \mathbb{Z}$, *the* $k$-*lower level set and* $k$-*upper level set of* $\sigma$ *are defined as*

$$\mathbb{L}_\sigma^k = \left\{ x \in \mathbb{X} : x \prec_b \sigma,\, dim(x) = k \right\}, \quad \mathbb{U}_\sigma^k = \left\{ x \in \mathbb{X} : \sigma \prec_b x,\, dim(x) = k \right\}. \tag{12}$$

Notice that, if $\dim(\sigma) = k$, then $\mathbb{L}_\sigma^{k'} = \emptyset$, $\forall k' \geq k$ and $\mathbb{U}_\sigma^{k'} = \emptyset$, $\forall k' \leq k$, due to monotonicity of dim function. Moreover, $\mathbb{L}_\sigma^{-1} = \mathbb{U}_\sigma^{K+1} = \emptyset$, which allows for compact representation as in equation 1.

For each $\mathbb{W}_k$, we can associate a signal called a $k$-cellular signal. The set of signals over $\mathbb{W}_k$ is denoted by $\Omega_k(\mathbb{W})$ for all $k = 0, 1, \ldots, K$. One can define a vector $\boldsymbol{x}_k \in \mathbb{R}^{N_k}$, whose entries are the signal values over $\mathbb{W}_k$, where $\boldsymbol{x}_k(\sigma)$ corresponds to the value of the signal on $\sigma$. Moreover, for some subset $\mathbb{S} \subseteq \mathbb{W}_k$, we write $\boldsymbol{x}_k(\mathbb{S})$ as the vector of values of its elements. If the signal is time-varying, we will denote the signal with matrix notation $\boldsymbol{X}_k \in \mathbb{R}^{N_k \times T}$ where $T$ is the total number of time instants and $\mathbf{x}_k^t$ denotes a $k$-cellular signal at time instant $t$. Time-lagged values of the signals corresponding to some vertices, edges, and polygons are shown in Figure 1.

We denote a collection of RKHS spaces that operate over $\Omega_k(\mathbb{W})$ as a connection RKHS, whose definition is given below:

**Definition 6** (Connection Reproducing Kernel Hilbert Spaces). *Let* $(\sigma, \tau)$ *be pairs of cells such that* $\dim(\sigma) = k$ *and* $\tau \in \mathbb{L}_\sigma^{k-1}$ *or* $\tau \in \mathbb{U}_\sigma^{k+1}$. *Without loss of generality, let us consider* $\tau \in \mathbb{U}_\sigma^{k+1}$. *A connection RKHS is the collection of RKHS over all the adjacent cell pairs* $(\sigma, \tau)$ *defined over the ACC, which is composed of* $2p$ *different kernel spaces for each* $(\sigma, \tau)$, *where* $p$ *is a natural number, namely* $\mathbb{F}_{\sigma,\tau}^p$ *and* $\mathbb{G}_{\sigma,\tau}^p$ *are given as*

$$\mathbb{F}_{\sigma,\tau}^p = \left\{ f_{\sigma,\tau}^p : \Omega_k(\mathbb{W}) \to \mathbb{R} \,\middle|\, f_{\sigma,\tau}^p(\boldsymbol{y}_k) = \sum_{t=p}^\infty \beta_{\sigma,\tau}^{p,t-p} \kappa_{\sigma,\tau}^p \left( \boldsymbol{y}_k, \boldsymbol{x}_k^{t-p}\left(\mathbb{L}_\tau^k\right) \right) \right\}, \tag{13}$$

$$\mathbb{G}_{\sigma,\tau}^p = \left\{ g_{\sigma,\tau}^p : \Omega_{k+1}(\mathbb{W}) \to \mathbb{R} \,\middle|\, g_{\sigma,\tau}^p(\boldsymbol{y}_{k+1}) = \sum_{t=p}^\infty \delta_{\sigma,\tau}^{p,t-p} \lambda_{\sigma,\tau}^p \left( \boldsymbol{y}_{k+1}, \boldsymbol{x}_{k+1}^{t-p}(\tau) \right) \right\}. \tag{14}$$

An example of an ACC with $K = 2$ is illustrated in Figure 5, which could represent for example a power grid network where 0-cells (nodes) are power stations, substations, and transformers, 1-cells

---

[1]The proposed method can be extended to higher-dimensional ACCs.

(edges) are transmission lines connecting them, and 2-cells (polygons) represent power distribution regions enclosed by transmission lines.

In Figure 5, the signals $\boldsymbol{x}_i(\sigma)$ are named with this convention, that is, in a power grid for example, $\boldsymbol{x}_0(v)$ can be the voltage at a junction $v$, $\boldsymbol{x}_1(e)$ can be the current flow through the line $e$ and $\boldsymbol{x}_2(\tau)$ can be total power requirement in enclosed area $\tau$. We also define $\Omega^{-1}(\mathbb{W}) = \Omega^{K+1}(\mathbb{W}) = \{0\}$ for simplicity of the representations and boundary maps $\boldsymbol{B}_k : \mathbb{R}^{N_k} \to \mathbb{R}^{N_{k-1}}$, satisfying $\boldsymbol{B}_k \boldsymbol{B}_{k+1} = 0$. A canonical way that accounts for the topology is to define boundary maps according to the boundary relation and the orientation given to the cells Sardellitti et al. (2021); Battiloro et al. (2023). Figure 5 shows the established orientations.

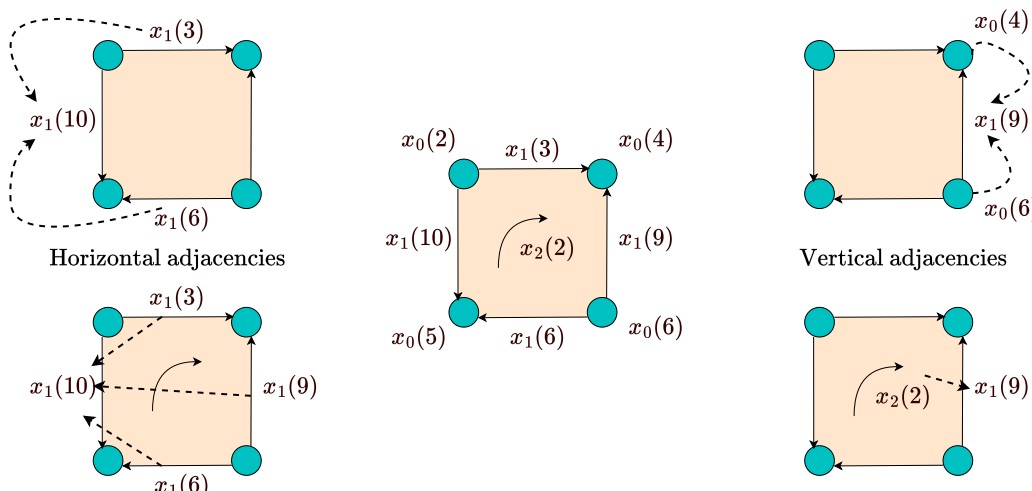

Figure 4: Consider the cell located in the middle. On the left-hand side, horizontal adjacencies are represented. The edge signal $\mathbf{x}_1(10)$ is horizontally adjacent to $\mathbf{x}_1(3)$ and $\mathbf{x}_1(6)$ through the corresponding shared nodes, and to $\mathbf{x}_1(3)$, $\mathbf{x}_1(9)$, and $\mathbf{x}_1(6)$ through polygon. On the right side, vertical adjacencies are illustrated. The edge signal $\mathbf{x}_1(9)$ is vertically adjacent to node signals $\mathbf{x}_0(4)$ and $\mathbf{x}_0(6)$, and to the shared polygon signal $\mathbf{x}_2(2)$. Similar horizontal and vertical adjacencies occur for other edge signals.

Here, we would like to point out that, in general, the time-series value defined over a particular cell depends on its neighbors both horizontally and vertically Krishnan et al. (2024). Horizontal dependency refers to the influence of neighboring cells with the same dimension, while vertical dependency captures the relationship between the data defined over a cell and the data associated with cells of the immediately lower and higher dimensions (see Figure 4) . For instance, consider a water distribution network modeled as an ACC, where data is defined on a specific pipe, represented as an edge in the complex. The horizontal dependency captures how the flow rate in this pipe is related to the flow rates in neighboring pipes, those connected through a common junction (node) or forming part of the same loop (polygon). These relationships characterize how changes in one part of the network can affect, and be affected by, flows in adjacent parts. In contrast, the vertical dependency describes interactions across different topological levels: namely, how the flow through a pipe relates to the pressures at the nodes it connects to, as well as the total circulating flow within the polygon to which it belongs. Together, these dependencies provide a rich, multi-scale representation of the physical and topological interactions within the network. Together, these dependencies allow us to model both local and global interactions in the network, leveraging the rich structure provided by the ACC representation. We denote $\mathbb{L}_c^k$ as the set of horizontal neighbors of cell $k$ through the cell $c$, when $c = k - 1$. Similarly, when $c = k + 1$, the set of horizontal dependencies is expressed as $\mathbb{U}_c^k$.

## A.2 CELLULAR FILTERING

Let $r_k = \text{rank}(\boldsymbol{B}_k)$, then each boundary map admits a Singular Value Decomposition (SVD) $\boldsymbol{B}_k = \boldsymbol{U}_k \Lambda_k \boldsymbol{V}_k^T$, where $\boldsymbol{U}_k \in \mathbb{R}^{N_{k-1} \times r_k}$, $\boldsymbol{V}_k \in \mathbb{R}^{N_k \times r_k}$ and $\Lambda_k \in \mathbb{R}^{r_k \times r_k}$. We will denote the set of

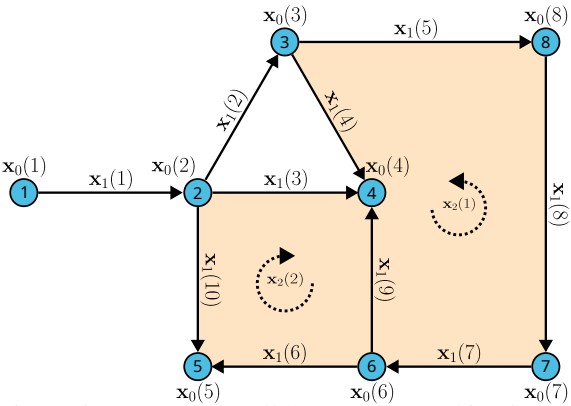

Figure 5: An example cellular complex and its signals

nonzero singular values of $\boldsymbol{B}_k$ as $\mathbb{B}_k$. Also, upper $\boldsymbol{L}_{k,1}$, lower $\boldsymbol{L}_{k,-1}$, and Hodge $k-$Laplacian $\boldsymbol{L}_k$ Sardellitti et al. (2021) are defined as:

$$\boldsymbol{L}_k = \boldsymbol{L}_{k,-1} + \boldsymbol{L}_{k,1} = \boldsymbol{B}_k^T \boldsymbol{B}_k + \boldsymbol{B}_{k+1} \boldsymbol{B}_{k+1}^T = \boldsymbol{R}_k \Phi_k \boldsymbol{R}_k^T, \tag{15}$$

where $\boldsymbol{R}_k \in \mathbb{R}^{N_k \times N_k}$ are the eigenvectors of the Laplacian matrix and $\Phi_k$ is a diagonal matrix whose elements are coming from the set $\mathbb{F}_k = \left\{\lambda^2 : \lambda \in \{0\} \cup \mathbb{B}_k \cup \mathbb{B}_{k+1}\right\}$ Sardellitti et al. (2021), respectively. The frequency response Sardellitti et al. (2021) of $\boldsymbol{f}_k \in \Omega^k(\mathbb{X})$ is given by

$$\hat{\boldsymbol{f}}_k = \boldsymbol{R}_k^T \boldsymbol{f}_k.$$

We can associate each element of $\hat{\boldsymbol{f}}_k$ with singular values of $\boldsymbol{B}_k$, $\boldsymbol{B}_{k+1}$ and the value 0, since the eigenvalues of the $k$-Laplacian is the squared singular values of $\boldsymbol{B}_k$, $\boldsymbol{B}_{k+1}$ and value 0. Therefore, for a frequency response $\hat{\boldsymbol{f}}_k$, $\hat{\boldsymbol{f}}_k(\lambda^2)$ means the frequency response at singular value $\lambda$. An input $k$-cell signal $\boldsymbol{f}_k$ can be filtered with a kernel $\hat{h}_k : \mathbb{R} \rightarrow \mathbb{R}$ as:

$$\boldsymbol{y}_k = \boldsymbol{R}_k \hat{h}_k(\Phi_k) \boldsymbol{R}_k^T \boldsymbol{f}_k = \boldsymbol{H}_k(\boldsymbol{L}_k) \boldsymbol{f}_k$$

where $\hat{h}_k(\Phi_k)$ is a diagonal matrix whose diagonal entries are the element-wise evaluations of $\hat{h}_k$ at $\mathbb{F}_k$. $\boldsymbol{H}_k(\boldsymbol{L}_k)$ is called as a $k$-cellular filter Isufi et al. (2025) which captures the spatial relationships between $k$-cell signals.

### A.3 NL-HORSO MODEL

In this section, we give algorithm and the extended schematic for the NL-HORSO model, which can be found in Figure 10 and Algorithm 1.

## B EXAMPLES OF MODELS UNDER THE NL-HORSO FRAMEWORK

This appendix will show the generalization capability of our algorithm. First we show how cellular convolution filters can be special case of our model then we explain how NL-HORSO generalizes graph learning methods using kernels. Finlay, we introduce RF-HORSO, a computationaly efficient version of NL-HORSO, and investigate its properties.

### B.1 LINEAR CELLULAR CONVOLUTION MODEL

Consider the simple filtering operation over a cellular complex defined as the matrix polynomial of the $k$-Hodge Laplacian as shown in equation 16.

$$\boldsymbol{y}_k = \boldsymbol{H}_k(\boldsymbol{L}_k) \boldsymbol{x}_k = \boldsymbol{H}_k(\boldsymbol{L}_{k,-1}) \boldsymbol{x}_k + \boldsymbol{H}_k(\boldsymbol{L}_{k,1}) \boldsymbol{x}_k \tag{16}$$

where the second equality is true due to $\boldsymbol{B}_{k+1} \boldsymbol{B}_k = 0$. For a finite cellular complex, any cellular filter can be written in terms of a finite ordered polynomial as shown in equation 17.

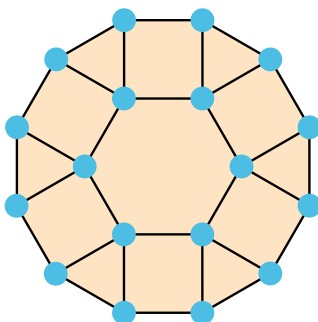
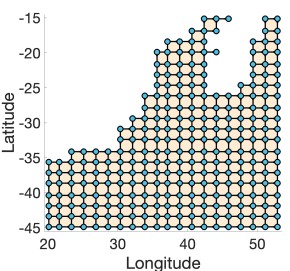
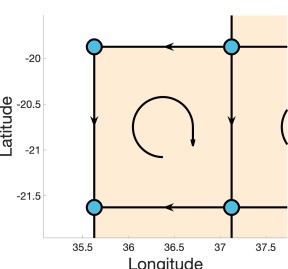

(a) Non-oriented cellular complex used for Synthetic Dataset.

(b) Oriented cellular complex used for Ocean Dataset.

(c) Orientation of Ocean Dataset.

Figure 6: Cellular complexes used in Synthetic and Ocean Datasets.

$$\boldsymbol{y}_k = \boldsymbol{H}_k\left(\boldsymbol{L}_{k,-1}\right)\boldsymbol{x}_k + \boldsymbol{H}_k\left(\boldsymbol{L}_{k,1}\right)\boldsymbol{x}_k = \sum_{q=0}^{K_l-1}\alpha_{k,-1,q}\boldsymbol{L}_{k,-1}^q\boldsymbol{x}_k + \sum_{q=0}^{K_u-1}\alpha_{k,1,q}\boldsymbol{L}_{k,1}^q\boldsymbol{x}_k \qquad (17)$$

Moreover, one may incorporate the functions that are related to the upper and lower adjacencies as

$$\boldsymbol{y}_k = \boldsymbol{B}_{k+1}\boldsymbol{H}_{k+1}\left(\boldsymbol{L}_{k+1}\right)\boldsymbol{x}_{k+1} = \boldsymbol{B}_{k+1}\sum_{q=0}^{K-1}\alpha_{k+1,q}\boldsymbol{L}_{k+1}^q\boldsymbol{x}_{k+1}. \qquad (18)$$

### B.1.1 KERNEL FUNCTIONS FOR LINEAR CELLULAR CONVOLUTION MODEL

Without loss of generality, let $\hat{h}_{k,1}$ and $\hat{h}_{k,-1}$ be positive functions over the eigenvalues of the Hodge-Laplacian. Moreover, let $\hat{h}_{k,1}, \hat{h}_{k,-1}$ are constant over $\{0\} \cup \mathbb{B}_k$ and $\{0\} \cup \mathbb{B}_{k+1}$, respectively. Notice that if the function includes a negative value, then it can be made positive simply by redefining the eigenvectors by their negatives. Given these functions, we can define upper and lower kernels related to $(\sigma, \sigma')$ as follows:

$$\kappa_{\sigma,\sigma',1}\left(\boldsymbol{x}_k, \boldsymbol{y}_k\right) = \sum_{n=1}^{N_k}\hat{h}(\phi_{k,n})\boldsymbol{r}_{k,n}[\sigma]\boldsymbol{y}_k[\sigma']\boldsymbol{x}_k[\sigma'],$$
$$\kappa_{\sigma,\sigma',-1}\left(\boldsymbol{x}_k, \boldsymbol{y}_k\right) = \sum_{n=1}^{N_k}\hat{h}(\phi_{k,n})\boldsymbol{r}_{k,n}[\sigma]\boldsymbol{y}_k[\sigma']\boldsymbol{x}_k[\sigma']. \qquad (19)$$

Typically $\boldsymbol{y}_k$ is given by the eigenvectors $\boldsymbol{r}_{k,n}$ of the Hodge-Laplacian $\boldsymbol{L}_k$, hence one gets the representation in equation 16. Moreover, due to Proposition 1, collection of RKHS created by equation 19 is a connection RKHS hence this is a subset of NL-HORSO model.

### B.1.2 NL-HORSO AS A GENERALIZATION OF LINEAR MODEL

Here, we want to show that a convolution operator can be written in terms of two operations: first, do a linear transformation to upper or lower dimensional cells, and second, come back to the original dimension with the boundary mapping as shown in equation 20.

$$\sum_{q=1}^{K_l}\alpha_{k,-1}\boldsymbol{L}_{k,-1}^q = \boldsymbol{B}_k^T\sum_{q=1}^{K_l}\alpha_{k,-1}\boldsymbol{B}_k\boldsymbol{L}_{k,-1}^{q-1} = \boldsymbol{B}_k^T\boldsymbol{F}_{k,-1}$$
$$\sum_{q=1}^{K_u}\alpha_{k,1}\boldsymbol{L}_{k,1}^q = \boldsymbol{B}_{k+1}\sum_{q=1}^{K_u}\alpha_{k,1}\boldsymbol{B}_{k+1}^T\boldsymbol{L}_{k,1}^{q-1} = \boldsymbol{B}_{k+1}\boldsymbol{F}_{k,1} \qquad (20)$$

which is applied to $\mathbf{x}_k \in \Omega^k(\mathbb{W})$. Here $\boldsymbol{F}_{k,1}, \boldsymbol{F}_{k,-1}$ are topology aware encoders that encode $k$-cell information onto the lower and upper dimensional cells and $\boldsymbol{B}_k, \boldsymbol{B}_{k+1}$ are the aggregators where the encoded information is merged according to the underlying topology.

A commutative diagram representing the operations is given in Figure 7, where $\boldsymbol{F}_{k,1}, \boldsymbol{F}_{k,-1}$ are mappings between the $k$-(co)chain[2] with its upper and lower (co)chains respectively, and $\boldsymbol{B}_{k+1}, \boldsymbol{B}_k$ are the boundary maps to get back to the original $k$-(co)chain space.

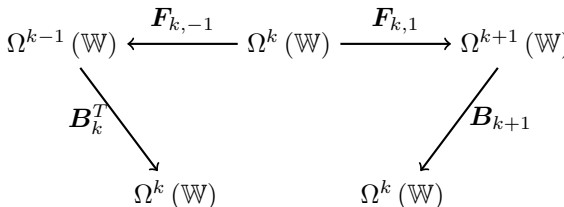

Figure 7: Functional diagram for the convolution operation over a cellular complex

For the linear model, the mappings $\boldsymbol{F} = \{\boldsymbol{F}_{k,-1}, \boldsymbol{F}_{k,1}\}$ determine two different properties of the information flow between $k$-cells of a structure. First, the degree of the polynomial shows how a $k$-cell is affected by its distant neighbors, and second, how important the information coming from each neighbor. In this way, $\boldsymbol{F}_{k,-1}$ and $\boldsymbol{F}_{k,1}$ determine the flow of information between adjacent cells connected by upper and lower-level cells within the topology. This is particularly important in equation 1, because $\boldsymbol{F}_{k,-1}$ and $\boldsymbol{F}_{k,1}$ allows simple and explainable parametrization of the information flow within the cell complex, thus allowing to learn how information is affected by the structure. This indicates that the convolution operation over a cellular complex can be thought of as encoding the information over upper and lower levels, and aggregating the encoded messages. This approach of convolution filtering can be described by NL-HORSO. The equation 17 is analogues to the functions $f_{\sigma,\rho}(.)$ and $f_{\sigma,\tau}(.)$ in equation 1. The arguments of these functions are the aggregated information from the lower and upper adjacent cells $\rho$ and $\tau$ of cell $\sigma$. This approach is particularly useful in various combinatorial models for regulating network flows. For example, the works Bodnar et al. (2022); Barbero et al. (2022) use a sheaf structure to address over-smoothing in Graph Neural Networks by regulating the flows.

Up to this point, functions arising from self-interactions of a (co)chain are investigated. However, most data on a cellular complex consists of layers with different modalities; for example, the electrical field of an electromagnetic wave can be represented as a vector field on the edges, while the magnetic field of the wave appears as a field over the polygons. In this case, one can project the magnetic field onto the edges with a linear model using the boundary maps. One example of such a linear model is shown in equation 18 for a general cellular network showing a projection to lower dimensional cells. Speaking of the same example, to capture the complex magnetic and electric field relationship over the cellular complex, a convolution operation is applied to the polygons allowing it to extract the information over polygons more efficiently. This can also be motivated by Maxwell's equations, where the relationship between the electric and magnetic fields is expressed through differential forms on the manifold. These relationships depend on the material properties of the medium, such as permittivity and permeability, which define the environment where the wave propagates He & Teixeira (2007). These properties can be seen as the parameters appearing in the polynomial expression in equation 18.

The diagram for equation 18 can be shown in Figure 7 but this approach is not helpful in this case because there are recursive operations of convolution. To avoid recursion, we use a single arrow for the linear transformation $\boldsymbol{B}_{k+1}\boldsymbol{H}_{k+1}(\boldsymbol{L}_{k+1})$ as shown in Figure 8. This arrow will contain all learnable parameters in the model, which are the polynomial coefficients of the filter. These parameters will adjust which information should be transferred to the vertical adjacent cells.

The operation in Figure 8 is analogues to function $g_{\sigma,\tau}(.)$ in equation 1, which encodes influence of signal defined over upper adjacent cell $\tau$ on the signal in cell $\sigma$. Similar mapping can be made for influence of signal defined over lower adjacent cell on signal in $\sigma$.

---

[2]A co-chain is a chain with boundary maps are reversed.

$$\Omega^k\left(\mathbb{W}\right) \xleftarrow{\quad \boldsymbol{B}_{k+1}\boldsymbol{H}_{k+1}\left(\boldsymbol{L}_{k+1}\right) \quad} \Omega^{k+1}\left(\mathbb{W}\right)$$

Figure 8: Functional diagram for the projection from $k+1$-cell to $k$-cell

## B.2 KERNEL MODELS OVER GRAPHS

Since the collection of cellular complexes incorporates the collection of graphs, graph-based models are a subset of the NL-HORSO model. For example, the NL-TISO Money et al. (2021) and RFNL-TIRSO models R. Money et al. (2023), are special cases of the NL-HORSO model. To show this, we assume a complete graph $\mathcal{G}$, and let $\mathbb{G}_{n,e}^p =$ and $\mathbb{F}_{n,e=(n,n')}^p = \mathcal{H}_{n'}^{(p)}$, where $\mathcal{H}_{n'}^{(p)}$ is as defined in Money et al. (2021); R. Money et al. (2023), then their collections constitute a connection RKHS. Thus, these kernel models are indeed a subset of the NL-HORSO model.

## B.3 RF-HORSO MODEL

The NL-HORSO method, as described in the main body of the paper, proposes the use of kernel methods to address nonlinearity. However, the general form of kernel methods applied to streaming data suffers from the problem of the curse of dimensionality. As the length of the time series increases, the number of features and parameters required to model the time series also increases. In the following section, we introduce RF-HORSO, which utilizes RF approximation to achieve a fixed-dimensional version of NL-HORSO. First we give a brief of how to approximate a function in RF space.

### B.3.1 RANDOM FEATURE APPROXIMATION

For this model, we assume that the function $f$ has the following form:

$$\forall \tau \in \mathbb{U}_\sigma^{k+1}, p \in \{1, 2, \ldots, P\}, \quad f_{\sigma,\tau}^p\left(\mathbf{x}_k^{t-p}(\mathbb{L}_\tau^k)\right) = \sum_{\sigma' \in \mathbb{L}_\tau^k} \frac{1}{\left|\mathbb{U}_\sigma^{k+1} \cap \mathbb{U}_{\sigma'}^{k+1}\right|} f_{\sigma,\sigma',1}^p\left(\mathbf{x}_k^{t-p}(\sigma')\right) \tag{21}$$

where $f_{\sigma,\sigma',1}^p$ is the function encoding the information from upper adjacencies and has the representation with kernel functions as in equation 2, where we express the function as weighted sum of kernel evaluations:

$$f_{\sigma,\sigma',1}^p(\boldsymbol{x}) = \sum_{t=p}^{T-p+1} \beta_{\sigma,\sigma',1,t}^p \kappa_{\sigma,\sigma',1}^p\left(\boldsymbol{x}, \boldsymbol{x}_k^{t-p}(\sigma')\right). \tag{22}$$

A normalization factor is placed in equation 21 in front of $f_{\sigma,\sigma',1}^p$ to minimize the recurrence of the same function. This form is also similar for the lower adjacency function $f_{\sigma,\rho}^p$ with reversed relationships and where the adjacency functions are denoted as $f_{\sigma,\sigma',-1}^p$. With these definitions, the resulting function space is a connection RKHS as seen as shown next:

**Proposition 1.** *Without loss of generality assume $\tau \in \mathbb{U}_\sigma^{k+1}$. The collection $\mathbb{S}$ of function spaces*

$$\mathbb{F}_{\sigma,\tau}^p = \left\{ f_{\sigma,\tau}^p \middle| f_{\sigma,\tau}^p\left(\mathbf{x}_k^{t-p}(\mathbb{L}_\tau^k)\right) = \sum_{\sigma' \in \mathbb{L}_\tau^k} \frac{1}{\left|\mathbb{U}_\sigma^{k+1} \cap \mathbb{U}_{\sigma'}^{k+1}\right|} f_{\sigma,\sigma',1}^p\left(\mathbf{x}_k^{t-p}(\sigma')\right), \, f_{\sigma,\sigma',1}^p \in \mathbb{F}_{\sigma,\sigma',1}^p \right\}$$

*constitutes a connection RKHS if each $\mathbb{F}_{\sigma,\sigma',1}^p$ are RKHS.*

*Proof.* By definition, $\mathbb{G}_{\sigma,\tau}^p$ already satisfies the RKHS condition. Proving that $\mathbb{F}_{\sigma,\tau}^p$ is an RKHS is trivial, as it is a widely studied RKHS over the direct sum of the respective kernel spaces Schölkopf & Smola (2002). Hence $\mathbb{S}$ is indeed a connection RKHS. $\qquad\square$

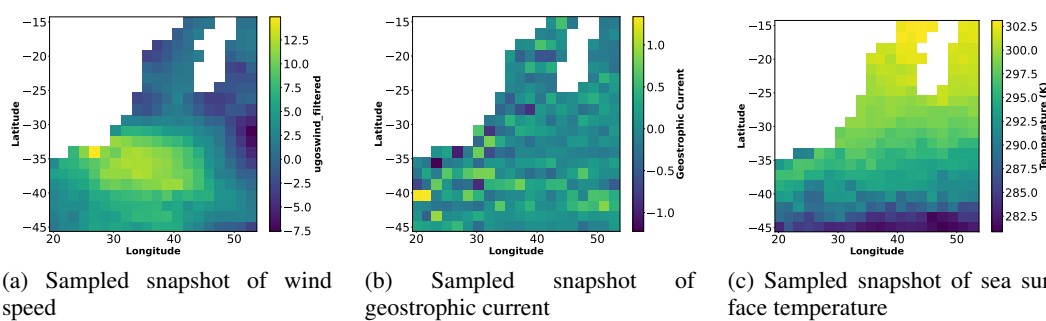

(a) Sampled snapshot of wind speed

(b) Sampled snapshot of geostrophic current

(c) Sampled snapshot of sea surface temperature

Figure 9: Snapshots of sampled versions of sea surface temperature, geostrophic current, and wind speed on 10 April 2023.

Now, let us assume that kernel functions are shift-invariant; therefore $\kappa(\boldsymbol{x}, \boldsymbol{y}) = \kappa(\boldsymbol{x} - \boldsymbol{y})$, where $\boldsymbol{x}, \boldsymbol{y} \in \mathbb{R}^d$. By Bochner's Theorem, there is a probability measure $\pi_\kappa$ such that

$$\kappa(\boldsymbol{x}) = \int_{\mathbb{R}^d} \exp(i\boldsymbol{v}^\top \boldsymbol{x}) d\pi_\kappa(\boldsymbol{v}) \tag{23}$$

Let $\boldsymbol{v} \in \mathbb{R}^{N_k}$ and $\kappa$ is a shift invariant kernel. The action of a matrix $\boldsymbol{S}_\mathbb{V} \in \mathbb{R}^{|\mathbb{V}| \times N_k}$ on the kernel function can be defined through this representation as

$$\forall \boldsymbol{x} \in \mathbb{R}^{|\mathbb{V}|}, \ \ (\boldsymbol{S}_\mathbb{V} \cdot \kappa)(\boldsymbol{x}) = \int_{\mathbb{R}^{|\mathbb{V}|}} \exp\left(i\boldsymbol{v}^\top \boldsymbol{x}\right) d\left(\boldsymbol{S}_\mathbb{V} \# \pi_\kappa\right)(\boldsymbol{v}) \tag{24}$$

where $\boldsymbol{S}_\mathbb{V} \# \pi_\kappa$ is the push-forward measure of $\pi_\kappa$ by $\boldsymbol{S}_\mathbb{V}$. Now let us consider the set of matrices that selects the entries of the vector $\boldsymbol{v}$ that correspond to a set $\mathbb{V}$. In order to simplify the expression, we use the same letter for every $\boldsymbol{S}_\mathbb{V} \cdot \kappa$ when it is evident from the context. Now, one can approximate each $\kappa$ by choosing $\{\boldsymbol{v}_d\}_{d=1}^D$ from some probability measure $\pi_\kappa$, then

$$\forall \boldsymbol{x} \in \mathbb{R}^{|\mathbb{V}|}, \ \ (\boldsymbol{S}_\mathbb{V} \cdot \kappa)(\boldsymbol{x}) \approx \frac{1}{D} \sum_{d=1}^D \exp\left(i\boldsymbol{v}_d^\top(\mathbb{V})\boldsymbol{x}\right) \tag{25}$$

where $\boldsymbol{v}_d(\mathbb{V})$ denotes the entries of $\boldsymbol{v}_d$ over the index set $\mathbb{V}$. Then we can write kernel function expressions under random feature approximation as follows

$$\begin{aligned} \kappa_{\sigma,\sigma',1}^p\left(\mathbf{x}_k^{t_1}(\sigma'), \mathbf{x}_k^{t_2}(\sigma')\right) &= \mathbf{z}_k^{t_1\top}\left(\sigma', \boldsymbol{v}_{k,1}\right)\mathbf{z}_k^{t_2}\left(\sigma', \boldsymbol{v}_{k,1}\right) \\ \lambda_{\sigma,\tau}^p\left(\mathbf{x}_{k+1}^{t_1}(\tau), \mathbf{x}_{k+1}^{t_2}(\tau)\right) &= \mathbf{z}_{k+1}^{t_1\top}\left(\tau, \boldsymbol{v}_{k+1}\right)\mathbf{z}_{k+1}^{t_2}\left(\tau, \boldsymbol{v}_{k+1}\right) \end{aligned} \tag{26}$$

where

$$\mathbf{z}_k^t(\mathbb{S}, \boldsymbol{v}) = \frac{1}{\sqrt{D}}\left[\sin(\boldsymbol{v}_1^\top(\mathbb{S})\mathbf{x}_k^t(\mathbb{S})) \ \ \ldots \ \ \cos(\boldsymbol{v}_D^\top(\mathbb{S})\mathbf{x}_k^t(\mathbb{S}))\right]^\top$$

and $\boldsymbol{v}_{k,1}$ denotes the random features corresponding to the upper adjacencies.

### B.3.2 ONLINE LEARNING OF RF-HORSO MODEL

Thus, we have the following approximations for upper adjacency transformations:

$$\begin{aligned} f_{\sigma,\sigma',1}^p\left(\mathbf{x}_k^{t-p}(\sigma')\right) &\approx \sum_{s=p}^{T-p+1} \beta_{\sigma,\sigma',1}^{p,s}\mathbf{z}_k^{s-p\top}\left(\sigma', \boldsymbol{v}_{k,1}\right)\mathbf{z}_k^{t-p}\left(\sigma', \boldsymbol{v}_{k,1}\right), \\ g_{\sigma,\tau}^p\left(\mathbf{x}_{k+1}^{t-p}(\tau)\right) &\approx \sum_{s=p}^{T-p+1} \delta_{\sigma,\tau}^{p,s}\mathbf{z}_{k+1}^{s-p\top}\left(\tau, \boldsymbol{v}_{k+1}\right)\mathbf{z}_{k+1}^{t-p}\left(\tau, \boldsymbol{v}_{k+1}\right). \end{aligned} \tag{27}$$

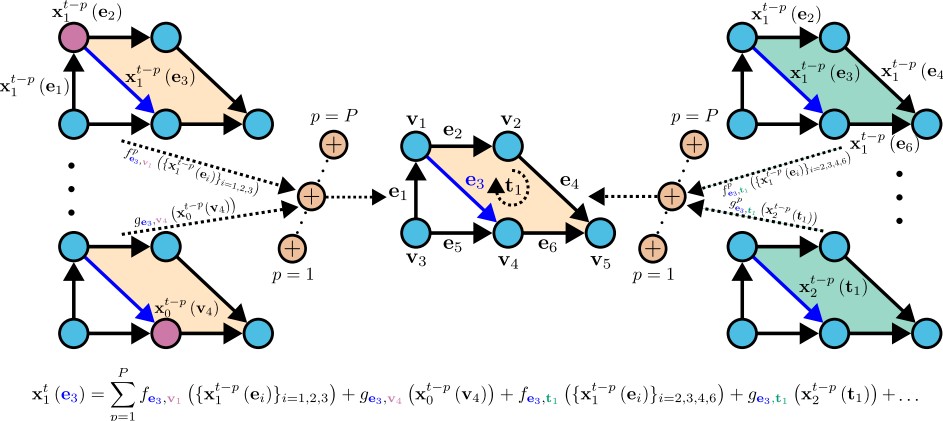

$$\mathbf{x}_1^t(\mathbf{e}_3) = \sum_{p=1}^{P} f_{\mathbf{e}_3,\mathbf{v}_1}\left(\{\mathbf{x}_1^{t-p}(\mathbf{e}_i)\}_{i=1,2,3}\right) + g_{\mathbf{e}_3,\mathbf{v}_4}\left(\mathbf{x}_0^{t-p}(\mathbf{v}_4)\right) + f_{\mathbf{e}_3,\mathbf{t}_1}\left(\{\mathbf{x}_1^{t-p}(\mathbf{e}_i)\}_{i=2,3,4,6}\right) + g_{\mathbf{e}_3,\mathbf{t}_1}\left(\mathbf{x}_2^{t-p}(\mathbf{t}_1)\right) + \dots$$

Figure 10: Consider an Abstract Cellular Complex (ACC) consisting of one polygon, four edges, and four nodes. The time-series value of the edge signal $\mathbf{x}_1^t(4)$ is expressed as the sum of nonlinear functions applied to the $P$ time-lagged values of: (i) neighboring edge signals connected through node 1, node 3, and polygon 1; (ii) node signals at nodes 1 and 3; and (iii) the polygon signal at polygon 1. Similarly, time-series signals defined over all entities in the ACC can be represented in this manner. Notably, each of these time series can be modeled independently, enabling the development of scalable learning algorithms.

As said, the expressions for lower adjacencies are similar. Then we define the following parameters

$$\mathbf{a}_{\sigma,\sigma',1}^p = \sum_{s=p}^{T-p+1} \beta_{\sigma,\sigma',1}^{p,s} \mathbf{z}_k^{s-p}\left(\sigma', \boldsymbol{v}_{k,1}\right),$$

$$\mathbf{c}_{\sigma,\tau}^p = \sum_{s=p}^{T-p+1} \delta_{\sigma,\tau}^{p,s} \mathbf{z}_{k+1}^{s-p}\left(\tau, \boldsymbol{v}_{k+1}\right), \tag{28}$$

which leads to the following representation of equation 27

$$f_{\sigma,\sigma',1}^p\left(\mathbf{x}_k^{t-p}(\sigma')\right) \approx \mathbf{a}_{\sigma,\sigma',1}^{p\top} \mathbf{z}_k^{t-p}\left(\sigma', \boldsymbol{v}_{k,1}\right),$$

$$g_{\sigma,\tau}^p\left(\mathbf{x}_{k+1}^{t-p}(\tau)\right) \approx \mathbf{c}_{\sigma,\tau}^{p\top} \mathbf{z}_{k+1}^{t-p}\left(\tau, \boldsymbol{v}_{k+1}\right). \tag{29}$$

The collection of the parameters using the notation in Section 3 can be defined as

$$\theta_{\sigma,\tau}^p = \left[\left\|_{\sigma' \in \mathbb{L}_\tau^k} \mathbf{a}_{\sigma,\sigma',1}^p\right\| \|\mathbf{c}_{\sigma,\tau}^p,\right.$$

where $\|$ is the aggregation operator, i.e., concatenates each vector in the expression to a larger vector. Since the number of weights is constant, i.e., does not depend on the episode length as in more general models, we also define $\tilde{\theta}_{\sigma,\tau}^p = \theta_{\sigma,\tau}^p$.

Moreover, to estimate the gradient vector $\boldsymbol{q}^t$, we do the following: First, rewrite the loss function $l_\sigma^t$ as

$$l_\sigma^t(\theta_\sigma) = \frac{1}{2}\mu \sum_{s=P}^{t} \gamma^{t-s}\left(\mathbf{x}_k^s[\sigma] - \mathbf{z}_\sigma^{s\top}\theta_\sigma\right)^2, \tag{30}$$

where $\mathbf{z}_\sigma^s$ is the concatenation of $\mathbf{z}_k^{s-p}\left(\sigma', \boldsymbol{v}_{k,1}\right)$ and $\mathbf{z}_{k+1}^{s-p}\left(\tau, \boldsymbol{v}_{k+1}\right)$ corresponding to cell $\sigma$. Hence, $l_\sigma^t(\theta_\sigma)$ can be written as

---

**Algorithm 1** NL-HORSO Algorithm

---
2

**Input:** Abstract Cellular Complex $\mathbb{W} = (\mathbb{X}, \dim, \prec_b)$, regularization parameter $\lambda_1, \lambda_{-1} > 0$, dimension $k$, number of realizations $T$, time lag $P$
**Generates:** Time-varying kernel coefficients $\theta[t]$, Autoregressive Process Estimates $\hat{x}_k[t]$
**Initialize:** $t \leftarrow 0, \theta[0] \leftarrow 0$
Construct $\{x^p\}_{p=0}^{P-1} \in \mathbb{R}^{P \times |\mathbb{W}_{k-1}, \mathbb{W}_k, \mathbb{W}_{k+1}|}$ with first $P$ observations
**for** t=P¡T **do**
    Observe $\{x^{t-p}\}_{p=1}^P$, $x^t$ over $\mathbb{W}$.
    **for** $p = 1$ **to** $P$ **and** $\sigma \in \dim^{-1}(\{k\})$ **do**
        Update $q_\sigma^p[t]$ according to the model (e.g. equation 2)
        Compute $\tilde{\theta}[t]$ according to the model
        **for** $c \in \mathbb{U}_\sigma^{k+1} \cup \mathbb{L}_\sigma^{k-1}$ **do**
            Compute $\theta_{\sigma,\tau}^p[t+1]$ with equation 6
        **end for**
    **end for**$\theta[t+1]$ $\hat{x}_k[t+1]$ according to equation 1
**end for**

---

$$l_\sigma^t(\theta_\sigma) = \frac{1}{2}\mu \sum_{s=P}^t \gamma^{t-s}\mathbf{x}_k^s[\sigma]^2 + \gamma^{t-s}\theta_\sigma^\top \mathbf{z}_\sigma^s \mathbf{z}_\sigma^{s\top}\theta_\sigma - 2\gamma^{t-s}\mathbf{z}_\sigma^{s\top}\theta_\sigma \mathbf{x}_k^s[\sigma]$$

$$= \frac{1}{2}\mu \sum_{s=P}^{t-1} \gamma^{t-s}\mathbf{x}_k^s[\sigma]^2 + \frac{1}{2}\theta_\sigma^\top \Phi_\sigma^t \theta_\sigma - \mathbf{r}_\sigma^{t\top}\theta_\sigma, \tag{31}$$

where

$$\Phi_\sigma^t = \mu \sum_{s=P}^t \gamma^{t-s}\mathbf{z}_\sigma^s \mathbf{z}_\sigma^{s\top}$$

$$\mathbf{r}_\sigma^t = \mu \sum_{s=P}^t \gamma^{t-s}\mathbf{z}_\sigma^s \mathbf{x}_k^s[\sigma] \tag{32}$$

These two matrices can be updated online as

$$\Phi_\sigma^t = \gamma\Phi_\sigma^{t-1} + \mu\mathbf{z}_\sigma^t \mathbf{z}_\sigma^{t\top}$$
$$\mathbf{r}_\sigma^t = \gamma\mathbf{r}_\sigma^{t-1} + \mu\mathbf{z}_\sigma^t \mathbf{x}_k^t[\sigma] \tag{33}$$

Hence, given these, the gradient vector corresponding to the coefficients of the cell $\sigma$, $q_\sigma^t$ is given as

$$q_\sigma^t = \Phi_\sigma^t \tilde{\theta}_\sigma^t - \mathbf{r}_\sigma^t \tag{34}$$

Then, equation 6 can be used to learn kernel parameters. In Algorithm 2, this process is summarized.

### B.3.3 DISCUSSION ON FREQUENCY-LIKE PARAMETRIZATION IN RF-HORSO

Recall that in Figure 7, the convolution operation was encoded in the functions $F_{k,1}, F_{k,-1}$ which manages the frequency content of the filtering operation achieved by $H(\boldsymbol{L}_{k,1}), H(\boldsymbol{L}_{k,-1})$ respectively. As shown in the Appendix B.1.1, these functions are related to kernels derived from these filtering operations. In this linear example, kernel functions are linear hence, it is evident how the frequency content is affected by the filtering operation. On the other hand, to achieve generalization capability NL-HORSO model "forgets" about the frequency and embeds this information into the functions $f_{\sigma,\tau}$. However, with the introduction of RF-HORSO, this frequency content is embedded into the cosine and sine functions and the random frequencies of the model. Therefore, in the

---

**Algorithm 2** RF-HORSO Algorithm

2

---

**Input:** Abstract Cellular Complex $\mathbb{W} = (\mathbb{X}, \dim, \prec_b)$, regularization parameter $\lambda_1, \lambda_{-1} > 0$, dimension $k$,number of realizations $T$, time lag $P$
**Generates:** Time-varying kernel coefficients $\breve{\theta}^t$, Autoregressive Process Estimates $\hat{x}_k^t$
**Initialize:** $t \leftarrow 0, \breve{\theta}^0 \leftarrow 0, \Phi^0 \leftarrow 0, \mathbf{r}^0 \leftarrow 0$
Construct $\{x^p\}_{p=0}^{P-1} \in \mathbb{R}^{P \times |\mathbb{W}_{k-1}, \mathbb{W}_k, \mathbb{W}_{k+1}|}$ with first $P$ observations
**for** t=P¡T **do**
$\quad$ Observe $\{x^{t-p}\}_{p=1}^P, x^t$ over $\mathbb{W}$.
$\quad \tilde{\theta}^t \leftarrow \breve{\theta}^t$
$\quad$ **for** $p = 1$ **to** $P$ **and** $\sigma \in \mathbb{W}_k$ **do**
$\quad\quad$ Update $\Phi_\sigma^t$ and $\mathbf{r}_\sigma^t$ with equation 33
$\quad\quad$ Update $q_\sigma^{p,t}$ according to the RF-HORSO model, see equation 34
$\quad\quad$ **for** $c \in \mathbb{U}_\sigma^{k+1} \cup \mathbb{L}_\sigma^{k-1}$ **do**
$\quad\quad\quad$ Compute $\tilde{\theta}_{\sigma,\tau}^{p,t+1}$ with equation 6
$\quad\quad$ **end for**
$\quad$ **end for** $\breve{\theta}^{t+1}$ $\hat{x}_k^{t+1}$ according to equation 1
**end for**

---

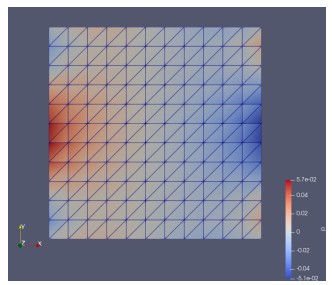

(a) Boundary condition of pressure field over unit square

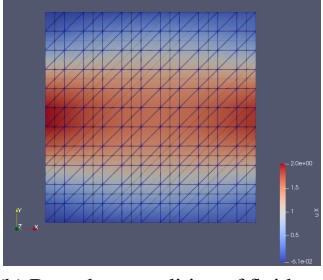

(b) Boundary condition of fluid velocity field on x direction

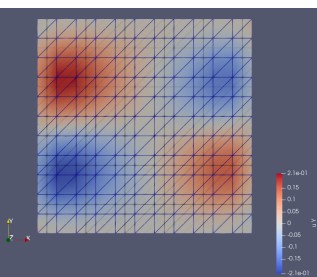

(c) Boundary condition of fluid velocity field on y direction

Figure 11: Snapshot of Navier-Stokes data with an overlay of cellular complex.

RF-HORSO model, there is a clear indication of what a frequency is, e.g., between the connected components $(\sigma, \sigma')$, we expect the change of information to be higher if their random frequencies are higher. Therefore, in the RF-HORSO model or in general, in a model induced by a shift-invariant kernel, the frequency information is encoded in the probability distributions related to each $(\sigma, \sigma')$ or $(\sigma, \tau)$. However, there is a trade-off; if the random frequency distributions are kept constant, then the expressive power of RF-HORSO is less, and if they are too diverse, then the model is vulnerable to overfitting and loss of geometric information coming from the model. This distinction is also made in the seminal paper Bodnar et al. (2022), where the authors solve this problem by learning diverse descriptions related to the connections. The RF-HORSO model may need such an approach; however, joint online learning of kernel parameters and random frequencies is limited in the literature Nguyen et al. (2017); Zhen et al. (2020), and we leave such extensions as future work. Learning of probability distributions themselves may open new avenues in terms of interpretability, expressiveness and flexibility of cellular complex based models.

### B.3.4 DYNAMIC REGRET ANALYSIS FOR RF-HORSO

In this section we give the details of the dynamic regret analysis in Section 4. Detailed descriptions of the Assumptions A1-A6 are given as follows:

- **A1: Bounded samples:** For all the time series samples, there exists $B_x > 0$ such that $\{\|x_i^t[\sigma]\|^2\}_{i \in \{k-1, k, k+1\}, \sigma \in \mathbb{W}_i, t} \leq B_x \leq \infty$.

- **A2: Shift-invariant kernels:** Kernels used are shift-invariant, i.e., $k(x_i, x_j) = k(x_i - x_j)$.

- **A3: Bounded minimum eigenvalue of $\Phi^t$:** There exists $\rho_l > 0$ such that $\Lambda_{\min}(\Phi^t) > \rho_l$, where $\Lambda_{\min}(\cdot)$ denotes the minimum eigenvalue, where

- **A4: Bounded maximum eigenvalue of $\Phi^t$:** There exists $L > 0$ such that $\Lambda_{\max}(\Phi^t) < L < \infty$, where $\Lambda_{\max}(\cdot)$ denotes the maximum eigenvalue.

- **A5: Lipschitz contunity of the cost function $h_\sigma^t$:** There exists $L_h \geq 0$ such that $\frac{\left|h_\sigma^t(\theta) - h_\sigma^t(\theta')\right|}{\|\theta - \theta'\|_2} \leq L_h$.

- **A6: Finite learning rate $a^t$:** $a^t := 1/L$.

Assumption **A1** is reasonable in practical scenarios, as signals encountered in real-world applications, such as those from sensors, financial systems, or physical processes, are naturally bounded due to physical or operational constraints. Assumption **A2** is satisfied by commonly used kernel functions, including the Gaussian and Laplacian kernels, where the induced feature maps ensure positive semi-definiteness and smooth variation, which are desirable properties for modeling. Assumption **A3** relies on the structure of the matrix $\Phi[t]$, which is constructed as a sum of rank-one matrices generated from feature vectors. This matrix will be of full rank as long as the feature vectors are linearly independent, an assumption that is generally satisfied when sufficient data diversity is present. This condition is crucial for ensuring the strong convexity of the loss function, which underpins the theoretical guarantees provided in the subsequent analysis. Assumption **A4** follows directly from **A1**, leveraging the fact that the trace of $\Phi[t]$, equal to the sum of its eigenvalues, is bounded due to the bounded nature of the signal. Assumption **A5** is to ensure that the cost function is regular, keeping the iterations more stable. Finally, assumption **A6** is made to achieve the maximum possible convergence speed without compromising the stability of the algorithm, as the constant $L$ is chosen to be bounded by the maximum eigenvalue of $\Phi[t]$. Together, these assumptions form a reasonable foundation for the theoretical development.

Consider a fixed episode length T. Then the dynamic regret arising from both COMID and random feature approximation is given by equation 9. Let us split this regret into two parts as

$$R_\sigma^{RF}[T] = R_\sigma[T] + \xi_\sigma[T] \tag{35}$$

where

$$R_\sigma[T] = \sum_{t=P}^{T-1} \left[ h_\sigma^t(\theta_\sigma^{t*}, \hat{\kappa}^t) - h_\sigma^t(\hat{\theta}_\sigma^t, \hat{\kappa}^t) \right]. \tag{36}$$

is the regret with respect to the optimal cost achieved by the random feature approximation, and

$$\xi_\sigma[T] = \sum_{t=P}^{T-1} \left[ h_\sigma^t(\theta_\sigma^t, \kappa^t) - h_\sigma^t(\theta_\sigma^{t*}, \hat{\kappa}^t) \right]. \tag{37}$$

is the regret caused by random feature approximation, where $\theta_\sigma^{t*}$ is the optimal parameter with respect to random feature approximation. Given these definitions, we give the proof of Theorem 1. We repeat it below for completeness.

**Theorem 2.** *Given the assumptions A1-A6, $\exists \epsilon, C > 0$ such that RF-HORSO satisfies the following bound for the regret:*

$$R_\sigma^{RF}[T] \leq \left( \left( 1 + \frac{L}{\rho_l} \right) \sqrt{2P \left( \left| \mathbb{L}_\sigma^{k-1} \cup \mathbb{U}_\sigma^{k+1} \right| + 2N_{k,\sigma} \right) DB_x} + (\lambda_{-1} + \lambda_1) \sqrt{P \left( \left| \mathbb{L}_\sigma^{k-1} \cup \mathbb{U}_\sigma^{k+1} \right| \right)} \right)$$
$$\times \left( \|\hat{\theta}_\sigma^{t-1*}[P]\|_2 + W_\sigma(T) \right) + \epsilon L_h T C. \tag{38}$$

*where $N_{k,\sigma} = \left| \bigcup_{\tau \in \mathbb{U}_\sigma^{k+1}} \mathbb{L}_\tau^k \cup \bigcup_{\rho \in \mathbb{L}_\sigma^{k-1}} \mathbb{U}_\rho^k \right|$.*

*Proof.* We will make use of the following theorems due to R. Money et al. (2023) to prove Theorem 2 and generalize the results for RFNL-TIRSO to RF-HORSO.

**Theorem 3.** *Under the assumptions of A1, A3, A4, and A6, the dynamic regret of RFNL-TIRSO with respect to the optimal cost function in the RF space satisfies*

$$R_n[T] \leq \left( \left( 1 + \frac{L}{\rho_l} \right) \sqrt{2PNDB_x} + \lambda \sqrt{PN} \right). \tag{39}$$

Theorem 3 states that the dynamic regret achieved by the RFNL-TIRSO algorithm is in $O\left(\sqrt{NP}\right)$, which is related to the number of nodes considered. Here we note that the first term stems from the number of random features $\mathbf{z}$ related to the node $n$, which is $2P\left(\left|\mathbb{L}_\sigma^{k-1} \cup \mathbb{U}_\sigma^{k+1}\right| + 2N_{k,\sigma}\right)D$ in RF-HORSO, and the second term comes from the number of elements inside the regularization term which is $P\left(\left|\mathbb{L}_\sigma^{k-1} \cup \mathbb{U}_\sigma^{k+1}\right|\right)$. And the inequality is applied differently for every $\lambda_{-1}$ and $\lambda_1$.

**Theorem 4.** *Under assumptions A1 and A2, there exists $\epsilon, C \geq 0$ such that the cumulative approximation error $\xi_n[T]$ of RFNL-TIRSO satisfies*

$$\xi_n(T) \leq \epsilon L_h TC, \tag{40}$$

which is similar for RF-HORSO due to being a random feature method, which yields the desired result.

Here, the first term on the right-hand side represents the tracking error, while the second term corresponds to the approximation error introduced by the RF approximation. Notice that if we set $\epsilon = \mathcal{O}\left(\frac{1}{\sqrt{T}}\right)$, the resulting dynamic regret becomes $\mathcal{O}(W_\sigma(T) + \sqrt{T})$. In this case, the dynamic regret is sublinear, provided that $W_\sigma(T)$ (cumulative path loss) is also sublinear. This requires some smoothness in the temporal variation of the signals. Here, $\epsilon$ denotes the maximum approximation error between the original kernel in the RKHS and its approximation in the random feature space. This error $\epsilon$ can be made arbitrarily small by increasing the number of random features, at the cost of increasing computational complexity.

$\square$

## C  EXPERIMENTATION

### C.1  SOUGHT DATASET AND BASELINE CHARACTERISTICS

**Dataset Choice**    Our framework imposes several structural requirements that limit the set of compatible benchmark datasets:

1. The data must be time-varying over a fixed complex or graph.

2. The data must be continuous-valued, as our random feature-based kernels rely on distances and inner products.

3. To evaluate generality, the signals must span multiple dimensions (e.g., nodes, edges, polygons).

Graph-based benchmark datasets are not directly suitable for our setting because they either provide only node-level signals, include categorical attributes, or rely on graph lifting. As stated in Section 2, our work assumes that data is defined over a fixed cellular complex with signals on multiple dimensions (e.g., nodes, edges, polygons). Consequently, such datasets do not meet the structural requirements of our framework

**Baseline Choice.**    We compare against a diverse set of baselines from both structured spatiotemporal modeling and general-purpose online learning:

- **SC-VAR** and **S-VAR**: Recent online autoregressive models defined over simplicial complexes. They serve as strong topological baselines that our model generalizes.

- **TIRSO**: A classical online sparse VAR method on unstructured vector-valued time series, providing a baseline for sparse temporal forecasting.

- **TopoLMS**: A linear online learning method over cellular complexes, enabling a direct comparison between linear and nonlinear learning in the same topological setting.

This selection enables systematic evaluation of higher-order structure, nonlinearity, and online learning capabilities across topology.

Table 2: tvNMSE values for different nonlinearity and dropout rates

|       | $d = 0.0$ | $d = 0.1$ | $d = 0.2$ | $d = 0.3$ | $d = 0.4$ | $d = 0.5$ | $d = 0.6$ | $d = 0.7$ |
|-------|-----------|-----------|-----------|-----------|-----------|-----------|-----------|-----------|
| $p_1$ | $0.068 \pm 0.003$ | $0.075 \pm 0.007$ | $0.073 \pm 0.004$ | $0.076 \pm 0.003$ | $0.082 \pm 0.004$ | $0.083 \pm 0.005$ | $0.078 \pm 0.006$ | $0.077 \pm 0.006$ |
| $p_2$ | $0.004 \pm 0.002$ | $0.005 \pm 0.002$ | $0.005 \pm 0.002$ | $0.005 \pm 0.002$ | $0.005 \pm 0.002$ | $0.005 \pm 0.002$ | $0.005 \pm 0.002$ | $0.005 \pm 0.002$ |
| $p_3$ | $0.238 \pm 0.013$ | $0.229 \pm 0.010$ | $0.244 \pm 0.017$ | $0.238 \pm 0.008$ | $0.241 \pm 0.013$ | $0.246 \pm 0.009$ | $0.246 \pm 0.015$ | $0.242 \pm 0.012$ |
| $p_4$ | $0.005 \pm 0.002$ | $0.005 \pm 0.002$ | $0.005 \pm 0.002$ | $0.005 \pm 0.002$ | $0.005 \pm 0.002$ | $0.005 \pm 0.002$ | $0.005 \pm 0.002$ | $0.005 \pm 0.002$ |

### C.2 DETAILS ON HYPERPARAMETER SEARCH

For real-world datasets, all algorithms are evaluated in a single-pass streaming setting, without re-training or access to future data. Hyperparameters were selected in advance based on preliminary experiments and held fixed throughout evaluation, eliminating variability due to random seeds or initialization.

RF-HORSO is empirically robust to hyperparameter selection and requires minimal tuning. Nevertheless, to ensure a fair comparison, we applied the same procedure to all baseline methods, selecting their best-performing hyperparameters under identical conditions.

### C.3 REPORTING OF EXPERIMENTATION

In this section, we test the sensitivity, performance, and additional capabilities by giving experimental details of the datasets and reporting additional experimental results on the RF-HORSO model. Codes for the model are available at [3]. The experimental results are reported with the metric tvNMSE $= \|z_k^t - \tilde{z}_k^t\|_2^2 / \|z_k^t\|_2^2$. tvNMSE is denoted as NMSE if it is the average tvNMSE over the samples. In all datasets, NMSE is reported as the average of the last 10 % of the run. In all of the experiments kernel probability distribution for the method is taken to be Gaussian with standard deviation parameter $\nu$.

### C.4 SYNTHETIC DATASET SPECIFICS AND ADDITIONAL RESULTS

In this section, the sensitivity of the RF-HORSO method for selected parameters polygon dropout rate $d$, the number of features $D$, standard deviation of the Gaussian distribution $\nu$. The dropout rate $d$ is the probability that a polygon is randomly dropped to obtain a new cellular complex structure. The extra experiment result for the dropout rate $d$ is given in Table **??**.

#### C.4.1 DATASET PROPERTIES

In all synthetic experiments, the cellular complex structure in Figure 6a has been used. This cellular complex includes different types of polygons capturing different numbers of multiple relationships, such as a triangle for a 3-way relationship, a square for a 4-way relationship, and finally a hexagon for a 6-way relationship. For the experiments, we used the following parameters while sweeping others:

Table 3: Common synthetic parameters used in the sensitivity analysis.

| | **Synthetic Dataset Parameters** | | | | | | | |
|---|---|---|---|---|---|---|---|---|
| | $d$ | $\nu$ | $D$ | $\lambda_{-1}$ | $\lambda_1$ | $P$ | Nonlinearity | $a^t$ |
| Nonlinearity Experiment (Figure 2a) | 0 | 1 | 5 | 0 | 0.01 | 4 | $p_3$ | $1/t$ |
| Bandwidth and Dropout Experiments (Figure 2c, Table 2) | 0.3 | 1 | 5 | 0.01 | 0.01 | 4 | $p_3$ | $1/t$ |
| AUC-ROC Experiment (Figure 2b, Table 1) | 0 | 1 | 5 | 1 | - | 4 | $p_5$ | $1/t$ |

Moreover, synthetic dataset at each $k$-cell is created by the linear combinations of functions residing on adjacent $k-1, k, k+1$-cells, namely $f_{\sigma,\sigma'}, g_{\sigma,\tau}, g_{\sigma,\rho}$ where these functions are given by nonlinearities. One distinction here is that for nonlinearities from $p_1$ to $p_4$, all connections uses the same nonlinearities, however, for $p_5$, upper connections uses $p_5$ while lower connections uses $p_5$ with $\nu_p = 1$. By using this nonlinearities, we generated $T = 1000$ timestep of data while letting $\mathbf{x}_{k+1}^t = \mathbf{x}_{k-1}^t = 0$.

---

[3] https://anonymous.4open.science/r/Higher-Order-Kernel-Approximation-D965

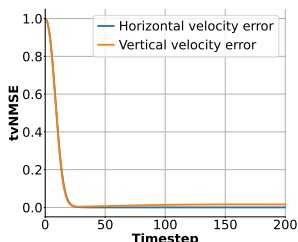

Figure 12: tvNMSE curve of Navier-Stokes Dataset

### C.5  CELLULAR COMPLEX BASED DATASETS AND ADDITIONAL RESULTS

#### C.5.1  NAVIER-STOKES DATASET

In this experiment, we tested our algorithm in convergence properties on a smooth dataset, which is inspired by the physical properties. Moreover, the Finite Element Method is run over the triangulation, which is basically a simplicial complex, which can be directly used in our experiments. The difference is that in finite element method is used to estimate functions over continuous surfaces Zienkiewicz et al. (2005), whereas our purpose is to attach a single value for each discrete element. To conduct this experiment, we made use of the Navier-Stokes example given by the FeniCS[4]Baratta et al. (2023), where the parametrization of the data is done as follows: Over a unit square, $T = 1000$ timesteps between 0-0.5 seconds are generated and grid of nodes placed $12 \times 12$ over the square were used to calculate the velocity and pressure fields. Triangulation made by the solver is used as the simplicial complex. Velocity fields are computed over the center of the edges, and the pressure field is calculated over the nodes. Moreover, the Reynolds number of the fluid is given as 250, and the polynomial degree for elements is given as 1, i.e., used linear elements. The boundary conditions at $t = 0$ and the simplicial complex used in the experiments are shown in Figure 11 within the ParaView Ahrens et al. (2005).

tvNMSE is reported for the estimation of horizontal and vertical velocity field from previous temperature and horizontal or vertical field data to show the convergence of the algorithm in Figure 12.

#### C.5.2  OCEAN DATASETS

In this experiment, the estimation performance of the RF-HORSO for different modalities over nodes, edges and polygons have been investigated over an oceanography dataset on the tip of South Africa. For this data, we investigated 2 cases: where the signals are only on edges which is named as Ocean Edge Dataset, and the signals are on nodes, edges, and polygons which is named as Ocean Cellular Dataset.

**Dataset Properties** The Ocean Dataset consists of daily average wind speed Copernicus Marine Service (2024), daily geostrophic current NOAA CoastWatch (2024a), and daily temperature over the sea surface on the tip of South Africa through 2023-2024 years, where a $20 \times 20$ grid is sampled over the area. The sampled value of each variable on 10 April 2023 is shown in Figure 9, Over this grid, an oriented cubical cellular complex is given as shown in Figures 6b, 6c. The total number of timesteps $T$ for 2023 is given as 365, and for 2024 is given as 361 days. Apart from the discussion in Appendix C.5.2, 2023 data is used in experiments.

The Ocean Edge Dataset consists of the daily geostrophic current gradient between sampled points and is based on the dataset introduced in Alain et al. (2023), and the Ocean Cellular Dataset consists of monthly average temperature over the nodes, wind speed gradient over the edges, and the geostrophic current curl over the squares. The process of getting the gradient and curl is detailed in Figure 13 and also detailed in Alain et al. (2023).

---

[4]To create the Navier-Stokes dataset, we made use of the official DOLFINx example: `https://docs.fenicsproject.org/dolfinx/v0.7.2/python/demos/demo_navier-stokes.html`.

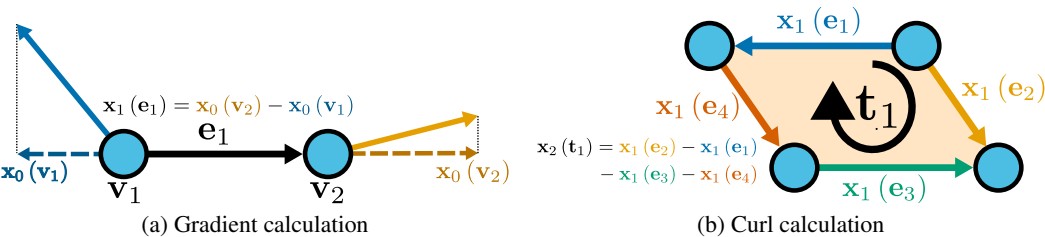

(a) Gradient calculation     (b) Curl calculation

Figure 13: Calculation of gradient and curl fields.

Since temperature is scalar, it is directly mapped to the sampled nodes, and vector fields are mapped to gradient and curl with the following: For any node, we project the vector in the direction of the edge and then add them up according to the orientation. Similarly, to calculate the curl, edge signals are summed up according to their compliance with its edge's orientation to the polygon's orientation. Therefore, in Figure 13b, $\mathbf{x}_1(\mathbf{e}_2)$ has a positive sign in contrast to the negative signs for the other edges, due to them having opposite orientation. The Ocean Cellular Dataset is first projected to the edges and then to the polygon by using this technique.

**Additional Results**   In Section 5, we hinted that TopoLMS learns the identity mapping. Here, we elaborate on this observation and present the learned coefficients on the Ocean Cellular Dataset. The coefficients for $t = 150$ and $t = 365$ are given in Table 4. As time progresses, TopoLMS converges toward the identity mapping, suggesting that the algorithm effectively ignores the cellular complex structure. Consequently, it fails to capture the intrinsic seasonal patterns present in the dataset, unlike RF-HORSO.

Table 4: Learned coefficients with TopoLMS at different time instants.

|         | $P = 2$, upper | $P = 1$, upper | $P = 0$ | $P = 2$, lower | $P = 1$, lower |
|---------|-------------|-------------|---------|-------------|-------------|
| $t = 150$ | $2.2 \times 10^{-4}$ | $-0.0019$ | **0.9982** | $3.17 \times 10^{-4}$ | $-0.0038$ |
| $t = 365$ | $2.53 \times 10^{-4}$ | $-4.93 \times 10^{-4}$ | **1.0008** | $1.81 \times 10^{-5}$ | $-0.0020$ |

This observation highlights a fundamental issue in cellular complex learning: determining when a dataset actually benefits from being modeled as a cellular complex, and conversely, when an algorithm leverages such a structure effectively. Some efforts aim to construct meaningful cellular complexes for given models Marinucci et al. (2025) and to provide benchmarks for topological datasets Telyatnikov et al. (2024), but these remain early steps. A systematic framework to evaluate when cellular complex methods provide added value is still lacking, and we leave this question for future work.

