# OpenReview forum: "Online Learning of Nonlinear Autoregressive Processes over Cellular Complexes"
_ICLR.cc/2026/Conference — Submitted to ICLR 2026_

### Official Review · Reviewer_rSv4 · 2025-10-29

**Soundness:** 3
**Presentation:** 2
**Contribution:** 2
**Rating:** 2
**Confidence:** 3

**Summary:**

The paper addresses the challenge of modeling high-dimensional, nonlinear, and non-stationary time-series data that possess higher-order structural dependencies (e.g., brain networks, fluid dynamics, water networks). The authors propose a novel nonlinear autoregressive framework defined over Abstract Cellular Complexes (ACCs), which is a generalization of graphs that includes nodes, edges, faces, etc. Each element (cell) can exchange information with its faces and cofaces via kernel-based interactions. The model is implemented in a Reproducing Kernel Hilbert Space (RKHS), capturing nonlinear dependencies while maintaining interpretability.

**Strengths:**

- This work is the first framework to study online nonlinear autoregressive modeling on cellular complexes.

- Proposed methods are mathematically rigor with multiple definitions and supported by theoretical error bound (theorem 1).

- Random Feature variant (RF-HORSO) makes the method tractable in streaming contexts.

- Experiments are performed in both synthetic and real-world datasets.

**Weaknesses:**

- I view the contribution on modeling the interaction between higher- and lower-order topologies is still novel and interesting. However, in a broader view, relationship between vertex to edge, edge to vertex, vertex-to-vertex, and edge-to-edge have been explored in nonlinear dynamical systems with autoregressive temporal dynamics [1,2], and online adaptation possibilities [3,4].

[1] Neural relational inference for interacting systems, ICML, 2018

[2] Neural Relational Inference with Efficient Message Passing Mechanisms, AAAI, 2021

[3] Online Relational Inference for Evolving Multi-agent Interacting Systems, NeurIPS, 2024

[4] Online Multi-agent Forecasting with Interpretable Collaborative Graph Neural Networks. IEEE TNNLS, 2022

- While conceptually and mathematically interesting, it would be nice if more real-world applications can be discussed where relationships between different-dimension cells emerged in high-order systems. While L33-36 discuss this a bit, I would like to see more mathematical properties or detailed examples.

- Experimental results are overall weak. Only ocean datasets are used (Ocean Cellular and Ocean Edge), and RF-HORSO performs better only in the Ocean Cellular dataset.

- RF-HORSO’s loss seems not converging in both datasets (Figure 3(a) and (b)). The lowest error is achieved at the very early timestep in Figure 3(a) and then the loss is highly fluctuating and increasing, which seems concerning. In contrast, other methods exhibit relatively stable though their loss is a bit higher.

**Questions:**

- Could the authors elaborate on why online algorithms are required for this benchmark (Ocean datasets)? What are the evolving factors that need to be adapted by models in this dataset? - otherwise, online learning may not be necessary.

- Why RF-HORSO exhibits much large variance in MSE over time?

- Are the number of learnable parameters comparable in the compared methods (RF-HORSO, S-VAR, SC-VAR)?

- Are these methods optimized with the similar level of learning rate?

- How fast are these methods in online optimization? - to make a fair comparison.

- Could the authors elaborate the definition of faces and cofaces before referring them for the readers?

- Probably, better to show the examples in Figure 1 better corresponding to the definitions. For example, L_sigma^k is not shown in Figure, instead showing L_tau^k. Also, it would be nice to elaborate the difference between blue-dot line and pink line in Figure 1 (though L127 discuss it a bit).

- I feel it's little confusing to follow the notation. In L144, x_k(S) denotes the values attached to k-cells, which I assume k-dimensional cells. That is, S includes the k-dimensional cells. In Figure 1, L_{tau}^{1} in x_{1}^{t-p}(L_{tau}^{1}) should represent 0-dimensional (not 1-dimensional, where k=1) neighbors of tau, based on Definition 2 (i.e., one dimension smaller), which is not aligned with L144.

---

> ### Author Response · Authors · 2025-11-21
> **Weaknesses Part 1**
>
> # Weaknesses
>
> ## Comparison with Graphical Models
>
> We agree that latent interaction modeling is a rich field, but there is a fundamental distinction in **Structure**, **Task**, and **Methodology** between the cited works and our proposal:
>
> The cited works ([1]-[4]) operate strictly on the **1-skeleton** (Graphs). They model interactions between agents (nodes) via latent edges. They **do not** model higher-order entities like **2-cells (polygons)**. In our framework, an edge signal is influenced not just by nodes, but by the enclosed 2-cell (e.g., circulation/curl driving edge flow). The methods in [1]-[4] are topologically blind to these "filled" structures. The primary task is **Latent Graph Inference**, observing node trajectories to infer a connection matrix. We address **Higher-Order Signal Processing**. We handle signals explicitly defined on nodes (e.g., scalar fields), edges (e.g., flow rates) and faces (e.g., curls). Our goal is to learn the nonlinear  functions ($f, g$) that govern the dynamics of these combinatorial signals using algebraic and combinatorial boundary maps.
>
> An important point is also to notice that  [3] employs **"Trajectory Mirroring"**, generating $2^n-1$ augmented trajectories by flipping axes, to mitigate model bias and ensure convergence, which includes exponential number of data augmentation. In contrast, our framework relies on the 1-skeleton of a **Cellular Complex** as a structural prior. This explicit topological inductive bias allows RF-HORSO to achieve theoretical dynamic regret bounds without requiring heuristic data augmentation to artificially inflate the sample size. Therefore, our work addresses the specific gap of **Topological Signal Processing**, utilizing algebraic and combinatorial tools distinct from the pairwise latent inference mechanisms in standard multi-agent literature.
>
> ## Real-World Applications
>
> We thank the reviewer for this constructive suggestion. In the revision, we will expand on the specific physical laws governing these inter-dimensional relationships to strengthen the motivation. We will explicitly map our topological operators ($f/g$) to the following interactions:
>
> 1. **Node-Edge Interaction (Navier-Stokes):** In our fluid dynamics experiments, data is explicitly defined on **nodes** (pressure field) and **edges** (velocity field). Standard graph signal models often treat edges merely as relational weights between nodes. Our framework models the physical coupling between the scalar pressure field (0-cells) and the vector velocity field (1-cells), effectively capturing the pressure gradient force that drives flow.
>
> 2.  **Edge-Polygon Interaction (Ocean Dynamics):** In our Ocean Cellular experiments, we model the interaction between **edges** (wind speed gradient) and **faces** (geostrophic current curl). This relationship is governed by Ekman transport (e.g., relating wind to ocean current transport, in our model it is the relation between 1 and 2-cells), which causally links the forcing terms defined on 1-cells to the vorticity response defined on 2-cells. The $g_{\sigma, \tau}$ operator in our model captures this specific inter-dimensional physical coupling, which pairwise graph models cannot natively represent.
>
> 3.  **Electromagnetism (Maxwell's Equations):** As noted in Appendix B.1.2, Maxwell's equations define Electric Fields as 1-forms (on edges) and Magnetic Flux as 2-forms (on faces). The temporal evolution is linked via Faraday's Law, a topological constraint relating the time derivative of the face signal to the curl of the edge signal.

---

> ### Author Response · Authors · 2025-11-21
> **Weaknesses Part 2**
>
> ## Experiments
>
> We respectfully wish to clarify the experimental scope and interpret the results in the context of online learning on non-stationary physical systems.
>
>  We clarify that the evaluation is **not limited to ocean datasets**.
>
>   * Synthetic Dataset: Used to validate sensitivity to nonlinearity and topological features (Section 5, Figure 2) .
>    * Navier-Stokes Dataset: A physics-based fluid dynamics simulation used to verify tracking of velocity/pressure fields (Section 5, Figure 12).
>    * Ocean Datasets: Used as the real-world benchmark consisting of 2 different datasets: Ocean Cellular and Ocean Edge.
>       * Ocean Cellular: Here, data exists on nodes, edges, and polygons. RF-HORSO significantly outperforms baselines because it utilizes the full hierarchy (vertical $g$ and horizontal $f$ operators) to model interactions between variables.
>       * Ocean Edge: Here, geostrophic currents are restricted to edges.
>
>    In online learning on non-stationary data, "fluctuating and increasing" loss is expected behavior during regime shifts, not a convergence failure. The error increase in Figure 3a corresponds to the transition into a more volatile physical regime. As noted in our response to Reviewer 1, the test period (2023-2024) coincides with record-breaking marine heatwaves and storm events (e.g., Cyclone Alvaro). The environment physically became harder to predict (higher variance in the true parameters).
>
>    Our dynamic regret bound (Eq. 11) explicitly scales with the path length $W_\sigma(T)$. When the physical system becomes more turbulent (the "storm"), $W_\sigma(T)$ increases, and the error bound naturally relaxes.

---

> ### Author Response · Authors · 2025-11-21
> **Questions**
>
> # Questions
>
> ## Question 1
> Online algorithms are necessary due to the **non-stationary** nature of ocean dynamics and the operational constraints of real-time monitoring.
>
> The system undergoes continuous shifts where static models fail. These include **Seasonal Transitions** (altering thermodynamic forcing) and **Extreme Events** (e.g., the 2024 record-breaking heatwaves and cyclones noted). An online algorithm is required to detect and adapt to these rapid changes in physical laws.
>
> In practical monitoring, data arrives in a continuous stream. Retraining a complex nonlinear model from scratch for every new sample is computationally prohibitive.
>
> ## Question 2
>
> We refer reviewer to check Weaknesses Part 2/Experiments.
>
> ## Question 3
>
> Although RF-HORSO has more learnable parameters per adjacency than S-VAR and SC-VAR, due to the use of random feature embeddings for nonlinear modeling, this does not translate into higher computational cost. RF-HORSO is cell-separable, meaning that each cell σ solves its own independent optimization problem based solely on its upper and lower adjacencies. As a result, the overall computation decomposes into many small subproblems that can be solved in parallel, and the per-problem complexity scales only with the local neighborhood size. In contrast, S-VAR and SC-VAR, despite having fewer parameters, require solving globally coupled optimization problems whose complexity grows with the size of the entire network, since their parameters are tied across all nodes/edges/faces. Thus, RF-HORSO offers a more scalable learning procedure: it may use richer parameterizations, but its decomposable structure yields lower effective computational complexity on large cellular complexes. The parameter number at each cell is actually represented in the full dynamic regret bound in Theorem 2 of Appendix B.3.4, where it is given as 2P (# of upper/lower/horizontal neighbors of \sigma) * D, where P is the time lag and D is the number of random features for each cell.
>
> ## Question 4
>
> For algorithms the  learning rate is not a hyperparameter, it is adapted  automatically as time progresses.  For example $a_t \in O(1/t)$ as shown in Table 3.
>
> ## Question 5
>
> RF-HORSO is designed for high-efficiency streaming.
>
> As noted in Section 3, the optimization decomposes into independent subproblems for each cell, allowing fully parallelized updates unlike coupled matrix-inversion methods.
>
> The update rule (Eq. 8) is a closed-form soft-thresholding operation. This requires only number of operations comparable to 2PD (#neighbors), per step, making it computationally comparable to linear filters and faster than retraining neural networks or standard kernel methods that scale with time.
>
> ## Question about faces and cofaces
>
> We thank the reviewer for pointing this out. Since it was a common definition on geometric machine learning we omitted their introduction. Definition in the introduction sections are informal definitions to introduce the subject in a friendlier way.
>
> ## Questions about notation
>
> Thanks for the feedback. We agree that in the definition instead of $L_{\sigma}$, $L_{\tau}$ should be used as $\sigma$ is used for the cells in the $k$-th layer and $\tau$ is used for the cells in the upper layers. The detailed definitions are found in the Appendix A.1 in Definition 5.
>
> However, we do not think there is a notation error. Since $\tau$ is 2-cell in the Figure, $L_{\tau}^1$ is showing the 1-cells that are attached to the $\tau$ so it is the set of 1-cells. The superscript of the notation shows the $k$-cells that are lower connected to the subscript $\tau$ in question.

---

### Official Review · Reviewer_C3kh · 2025-11-06

**Soundness:** 2
**Presentation:** 2
**Contribution:** 2
**Rating:** 2
**Confidence:** 2

**Summary:**

This paper proposes NL-HORSO for online learning of nonlinear autoregressive time series over Abstract Cellular Complexes (ACCs). The method uses kernel functions in RKHS to model dependencies between cells at different topological levels (nodes, edges, polygons), with RF-HORSO providing a scalable random feature approximation. Dynamic regret bounds are provided, and experiments are conducted on synthetic and real datasets.

**Strengths:**

1. Theoretically grounded: Provides dynamic regret bounds for online learning over topological structures using standard COMID analysis

2. Computationally tractable: RF-HORSO addresses scalability through random feature approximation with fixed dimensionality

3. General framework: Unified treatment of signals across nodes, edges, and higher-dimensional cells

4. Comprehensive appendix: Detailed proofs, derivations, and experimental settings

**Weaknesses:**

**Soundness Issues:**

1. Shallow theoretical contribution: Dynamic regret analysis (Theorem 1) applies standard COMID+RF results; extension to ACCs is primarily notational

2. Unvalidated assumptions: Assumptions A1-A6 are not verified empirically; practical implications of regret bound unclear given noisy experimental results

3. Missing justification: No explanation for why RKHS formulation is necessary or how it fundamentally connects to topological structure

**Presentation Issues:**

4. Unclear motivation flow: Abrupt jump from ACC motivation (Section 1) to RKHS formulation (Eq. 1) without explaining why kernels are needed

5. Heavy notation without intuition: Definitions 1-6 introduce dense formalism before providing insight; roles of f^p_{σ,c} vs g^p_{σ,c} unclear

6. Key insights buried: Connection to linear convolution (Appendix B.1) should appear in main text to build intuition

7. Late visualization: Figures 4-6 can help but appear after complex equations; early intuitive diagrams needed

**Contribution Issues:**

8. Incremental novelty: Primarily extends existing graph kernel methods (RFNL-TIRSO) to track upper/lower adjacencies; no new algorithmic or theoretical insights

9. Unconvincing experiments:

RF-HORSO shows high variance (Fig 3a) yet claims "competitive performance"

TopoLMS learns identity mapping (Table 4), suggesting topology isn't beneficial

No ablation comparing ACC vs. graph vs. independent time series


**Questionable datasets:**

Ocean/Navier-Stokes choices not justified - unclear why these require cellular complex structure rather than graphs


**Missing validation:**

No demonstration of when ACC structure provides advantage; claim of "first work on online learning over cellular complexes" lacks evidence this is actually a useful problem

**Questions:**

1. Can you provide intuition for when ACC structure provides advantage over graphs? What properties of the time series/system necessitate modeling triangles, not just edges?

2. Why does TopoLMS converge to identity (Table 4)? Does this suggest the cellular complex structure isn't beneficial for this data?
In Figure 3a, RF-HORSO shows much higher variance than baselines. Is this acceptable? How do you interpret "competitive performance" given this instability?

3. Can you compare against simply treating all cells as independent time series (ignoring topology)? This would clarify the value of the topological inductive bias.

4. The regret bound scales with path length W_σ[T]. How does this compare empirically to the observed error? Are the assumptions A1-A6 verified on your datasets?

---

> ### Author Response · Authors · 2025-11-21
> **Soundness Part 1/2**
>
> We reply the concerns of the reviewer in separate comments. This is the part 1 of the soundness discussion. References are appended at the end of the comment chain.
>
> # Soundness
> ## Theoretical contribution
> We thank the reviewer for this detailed observation. We agree that the **proof technique** for Theorem 1 adapts existing results from graph-based kernel learning (specifically adapting the results of *Money et al., 2023*). Because we formulate the problem in a Reproducing Kernel Hilbert Space (RKHS), the optimization guarantees naturally follow established kernel learning bounds.  One significant advantage of the proposed model is scalability which can be shown in terms of regret bound. In the graph based case the bound gets bigger as the total number of nodes increases. But in our case we model the system in a decomposable way where only local interactions through boundary maps matter, which means even if the original system has a very high number of nodes (cells) the per node bound will not grow with the total number of nodes.
>
> ## Assumptions
> We respectfully point out that Assumptions A1–A6 are standard theoretical conditions required for dynamic regret analysis in online learning. We did not perform "empirical verification" for them because they are either **inherent physical properties** of the data or **design choices** we explicitly control in the algorithm.Detailed justifications for these assumptions are provided in **Appendix B.3.4**.
>
> We address the specific reasoning for each assumption below:
>
> * **A1 (Boundedness) & A4 (Eigenvalue Bound):** A1 states that signals have finite power. In physical systems (like our Navier-Stokes and Ocean datasets), variables such as flow velocity or temperature are naturally bounded by finite energy constraints. An unbounded signal would imply physically impossible phenomena (infinite energy density). A4 is a direct mathematical consequence of A1.
> * **A2 (Shift-Invariant Kernels):** This is a **model design choice**, not an empirical hypothesis. We explicitly utilized Gaussian kernels (which are shift-invariant) to allow for the Random Fourier Feature approximation. While one could use non-shift-invariant kernels, this would require complex online updates for feature frequency distributions; an area where literature is sparse and often relies on computationally expensive combinatorial selection (Nguyen et al., 2017; Zhen et al., 2020). Our choice balances expressiveness with computational efficiency. Furthermore, **Figure 2c** demonstrates that the model is robust to the bandwidth parameter selection.
> * **A3 (Minimum Eigenvalue):** This condition requires the feature vectors to be linearly independent, which is generally satisfied when the data possesses sufficient diversity. This is a standard requirement essential for theoretical convergence rates.
> * **A5 (Lipschitz Continuity/Smoothness):** This assumption relates to the smoothness of the cost function. We empirically validated the implications of this assumption in **Figure 2a**. As predicted by the theory, when the underlying function is highly non-smooth (e.g., the high-frequency components in $p_3$ vs. the smoother $p_2$), the regret bound loosens, and the tracking error (tvNMSE) fluctuates more. The "noise" in the results is not an artifact but a validation that the algorithm behaves exactly as the smoothness assumption predicts.
> * **A6 (Step Size):** This is a hyperparameter we control. As shown in **Table 3** (Appendix C.4.1), we explicitly set the learning rate decay to $\mathcal{O}(1/t)$, which satisfies the theoretical requirement by construction.

---

> > ### Author Response · Authors · 2025-11-21
> > **Soundness Part 2/2**
> >
> > ## Justification
> > We respectfully disagree that the motivation is missing or that the connection to topology is unexplained. We address these points by clarifying the logical flow presented in the Introduction and the mathematical connections detailed in the Appendices.
> > The reviewer suggests an abrupt jump to Eq. 1. However, the motivation is explicitly built in the **Introduction (Section 1.1)** by identifying the gap between existing methods:
> > * **Hodge Laplacian models** are theoretically grounded but limited to **linear** relationships.
> > **Neural Architectures (GNNs/TNNs)** capture nonlinearity but are computationally heavy in streaming settings. To handle non-stationarity, they typically require complex continual learning mechanisms (e.g., experience replay buffers) and extensive retraining. In contrast, our kernel approach is **lightweight**, requiring storage of only the latest $P+1$ instants, while still providing theoretical tracking guarantees.
> >
> > We chose Kernels because they act as the necessary bridge: they are **universal approximators** (capturing nonlinearity) that enables **online learning with regret guarantees**. Unlike "black-box" neural networks, kernels allow us to attribute performance shifts to data characteristics (e.g., smoothness, as discussed in our response to Weakness 2).
> > The RKHS formulation is not independent of the topology; it is a direct nonlinear generalization of topological signal processing: As shown in **Appendix B.1.2**, our kernel formulation generalizes linear cellular convolutions. The kernel functions explicitly model the transport of signals across specific topological adjacencies (upper and lower).
> > The connection to topology is empirically verified in our **AUC-ROC experiments (Figure 2b, Table 1)**. By simply thresholding the learned kernel parameters, our model recovers the underlying topological structure (holes/adjacencies) from the data. This demonstrates that the learned kernels are topologically semantic, not just arbitrary weights.
> > Similar to (Bodnar et al., 2022), our kernels quantify how signals transform across cells, creating an explainable transport structure. As noted in **Appendix B.3.3**, we also identify that the frequency distributions generating the random features mathematically quantify how a signal changes over a connection, which we frame as a promising direction for future research.

---

> > > ### Author Response · Authors · 2025-11-21
> > > **Presentation**
> > >
> > > # Presentation
> > > ## Motivation Flow
> > > We respectfully disagree that the motivation is missing. The logical flow is explicitly built in the **Introduction (Section 1.1)** by identifying the specific gap between existing methods, and repeated in Justification section.
> > >
> > >
> > > ## Heavy Notation
> > > We thank the reviewer for the feedback on heavy notation. We acknowledge that the formalism required for cellular complexes should be dense. However, we respectfully point out that the intuitive roles of these functions are explicitly defined immediately following the definitions in **Section 2 (Lines 186-194)** and illustrated in **Figure 1**.
> > >
> > > To clarify the distinction explicitly:
> > > * **$f^p_{\sigma,c}(\cdot)$ (Horizontal Adjacency):** This function captures **"self-transformations"** or relationships between cells of the *same* dimension (e.g., edge-to-edge dependencies). For example, how the current in an edge is affected by signals (e.g. currents) on neighboring edges (connected via a common node $c$).
> > > * **$g^p_{\sigma,c}(\cdot)$ (Vertical Adjacency):** This function captures **"inter-dimensional transformations."** It models the relationship between the signal at cell $\sigma$ and the signal at a vertically adjacent cell $c$ of a *different* dimension (e.g., edge-to-node dependencies). For example, how the current in an edge is affected by the signal (e.g. temperature) at its endpoint node $c$. **Figure 1** provides a visual guide to these interactions.
> > >
> > > ## Key insights and Late Visualization
> > > We thank the reviewer for identifying this structural issue. We agree that the theoretical connection to linear convolution (Appendix B.1) is essential for intuition and was unfortunately displaced due to page constraints. In a future revision, we will integrate the theoretical discussion from Appendix B.1 directly into Section 2. This will immediately ground our nonlinear model as a direct generalization of standard spectral filters. Additionally, we will move the schematic diagrams (e.g., **Figure 10**) to the methodology section to provide immediate visual aid alongside the formal definitions.

---

> ### Author Response · Authors · 2025-11-21
> **Contribution**
>
> # Contribution
> ## Novelty
> We respectfully disagree that this is a trivial extension. We utilize **Cellular Complexes** to formulate the kernel model, where the distinction between **vertical** ($g$) and **horizontal** ($f$) adjacencies is central to the physical definition of the process.
>
> As shown in **Appendix B.1.2** (which we will move to the main text), this formulation is not just an application of kernels to a new domain; it is a **nonlinear generalization** of standard algebraic topology operators. It recovers linear cellular convolution as a special case while enabling the modeling of nonlinear interactions that standard graph models cannot natively represent.
>
> ## Experimentation
>
> ### High variance
> We argue that this variance is physically meaningful and reflects the model's response to rapid changes in the environment. This is hinted in Theorem 1, considering that the algorithm performance is dependent on the path length of the parameter, i.e. non-stationarity. This is inline with the weather events, and seasonal changes, where we see peak on the error curve. We cross-referenced the huge error spikes in 2024 section of **Figure 3c** with meteorological records:
>
> 1.  **Early 2024 Peak (Jan):** This aligns with the **impact of Tropical Cyclone Alvaro** in Madagascar on Jan 1, 2024 (Down to Earth., 2024). This event introduced immediate turbulence at the start of 2024. Our model attempted to track these sudden changes (resulting in the initial error spike).
> 2.  **Mid-2024 Peak (June):** This corresponds to the **South African Cut-Off Low storm complex** (June 1–5, 2024) (South African Weather Service Event Log., 2024). This rare event caused severe storm surges in the exact region of our dataset.
> 3.  **2024 vs. 2023 Dynamics:** Reports confirm that 2024 was the warmest year on record (World Meteorological Organization, 2025), with marine heatwaves in the region surpassing 2023 levels (CarbonCopy, 2025). This confirms that the dynamics in 2024 were distinct; the temperatures and flows were physically different from what the model learned in 2023. The variance we see is the model adapting to these new conditions. On the other hand baselines does not show this level of adaptation, where their behaviour is noisy around a point.
>
> ### Structure Insufficiency
> The fact that TopoLMS collapses to the Identity Mapping ($W \approx I$) is a not an indication of the topological structure.
>
> TopoLMS converges to the identity operator because it is a purely linear topology-aware LMS algorithm with very limited expressive capacity. Its updates rely only on horizontal adjacencies and instantaneous gradient corrections, so when exposed to nonlinear, cross-dimensional ACC dynamics, the model cannot represent the true interactions. In this situation, the least-squares optimum collapses to the trivial identity mapping , which is the best linear prediction available. Furthermore, TopoLMS cannot model the bidirectional relationships that naturally arise on cellular complexes, e.g., node signals influencing edge signals and edge signals influencing node signals. Instead, it learns either node-level or edge-level dependencies in isolation, preventing it from capturing the vertical (cross-dimensional) interactions that drive processes such as circulation, flux, or temperature, current coupling in our datasets. This collapse does not imply that the cellular complex structure is unhelpful. Rather, it demonstrates that linearity is insufficient to exploit the structure. RF-HORSO uses the same ACC topology but equips it with a nonlinear RKHS model, enabling it to learn rich, multiway, cross-dimensional dynamics.
>
> ### Ablation
> We agree with reviewer that we should have added independent time series. We could not incorporate RFNL-TIRSO because the network we considered is large (approx. 1000 different cells), and the method did not scale well.

---

> > ### Author Response · Authors · 2025-11-21
> > **Questions**
> >
> > # Questions
> >
> > ## Answer to Question 1
> > The ACC structure provides a decisive advantage when the data is defined on non-triangular cells or requires higher-order modeling beyond node signals.
> > In our **Ocean Dataset**, the underlying structure is a **Cubical Complex** (comprising squares, not triangles). Existing baselines like **S-VAR** and **SC-VAR** are restricted to Simplicial Complexes; consequently, on this dataset, they mathematically degenerate to standard graph models because they cannot process the square cells. Our ACC framework naturally handles these polygonal structures, allowing us to model the flow circulation (curl) over the squares, which S-VAR ignores.
> > In many physical systems (e.g., **Navier-Stokes**), data is explicitly defined over **edges** (velocity) and **nodes** (pressure). Standard graph kernel models are designed for node signals and cannot natively process edge-supported data without complex, ad-hoc transformations (e.g., line graphs). Our framework generalizes the kernel definition to any cell dimension, providing a natural and rigorous way to model these edge-based processes. In the ocean dataset under consideration, temperature measurements are naturally defined on 0-cells (nodes), while water currents correspond to 1-cell signals (edges). Physically, the wind flowing along an edge is driven by temperature and pressure differences between its endpoint nodes. These bidirectional dependencies inherently involve multiway interactions: an edge interacts with the wind flow on its incident edges, and those flows, in turn, depend on the temperature distribution over the nodes and the geostrophic current across the region enclosed by adjacent polygons. Such relationships cannot be faithfully captured by models defined purely on graphs, which encode only pairwise dependencies. Instead, they require a framework capable of handling higher-order structural interactions. Abstract Cellular Complexes (ACCs) provide exactly this: signals may be defined on nodes, edges, and polygonal faces, and the adjacencies between these cells, both horizontal (within the same dimension) and vertical (across dimensions), are explicitly encoded. In this setting, the coupled dynamics are naturally represented by the nonlinear, multiway functional dependencies formalized in our model (see Equation (1) and Figure 10). The ACC representation thus serves as a principled higher-order structure that captures the true physical interactions driving the ocean process, enabling expressive nonlinear autoregressive modeling.
> >
> > ## Question 2
> > We kindly refer reviewer to Structure Inefficiency section under Contribution comment.
> >
> > ## Question 3
> > This behavior is in fact desirable: an online algorithm should exhibit sensitivity to distributional shifts, because detecting and adapting to new regimes is precisely the hallmark of effective online learning. RF-HORSO increases its error only transiently during adaptation, whereas the baselines maintain low-variance errors simply because they fail to recognize that the underlying data distribution has changed. The detailed information can be found in High Variance section under Contribution comment.
> >
> > ## Question 4
> > We acknowledge that we did not include a specific "Independent Kernel" baseline in this study. Our primary focus was benchmarking against structure-aware methods (S-VAR, SC-VAR) to demonstrate the specific value of the nonlinear ACC formulation over existing topological approaches. We agree that isolating the "Self-Nonlinearity" contribution is a valuable ablation for future revisions.
> >
> > ## Question 5
> > The correlation between the error spikes in Figure 3c and the physical instability of the 2024 storm season validates the theoretical bound's dependence on path length ($W_\sigma[T]$). We refer reviewer to High Variance section under Contribution for the detailed discussion.
> >
> > As discussed in Assumptions under Soundness Section, Assumptions A1-A6 are design choices or inherent physical properties (boundedness), not hypotheses requiring statistical verification.

---

> > > ### Author Response · Authors · 2025-11-21
> > > **References**
> > >
> > > # References
> > >
> > > Yang et al. (2022) (Simplicial Convolutional Neural Networks)
> > >
> > > Isufi et al. (2025) (Topological Signal Processing and Learning)
> > >
> > > Ebli et al. (2020) (Simplicial Neural Networks)
> > >
> > > Bodnar et al. (2021) (Weisfeiler and Lehman Go Topological: Message Passing Simplicial Networks)
> > >
> > > Hajij et al. (2020) (Cell Complex Neural Networks)
> > >
> > > Battiloro et al. (2024) (E(n) Equivariant Topological Neural Networks)
> > >
> > > Bodnar et al. (2022) (Neural Sheaf Diffusion)
> > >
> > > Down to Earth. (2024) (Alvaro, first cyclone of 2024, hits Madagascar)
> > >
> > > South African Weather Service Event Log. (2024) (June 2024 South African storm complex)
> > >
> > > Carbon Copy (2025) (Record-breaking marine heatwaves devastated oceans over past 2 years)
> > >
> > > World Meteorological Organization (2025) (WMO confirms 2024 as warmest year on record)

---

### Meta-Review · Area_Chair_gXHd · 2026-01-07

**Summary:**

The authors introduce a framework for nonlinear autoregressive modeling over Abstract Cellular Complexes (ACCs) where predictive functions are defined in a reproducing kernel Hilbert space (RKHS) induced by shift-invariant kernels. In addition, the authors propose an efficient online learning algorithm to estimate these functions.

The main concerns raised are as follows:
1. Motivation is unclear.
2. Theoretical contribution is limited and unvalidated assumptions.
3. Experimental results are weak: more datasets are needed for the approach illustration.

**Reviewer Concerns:**

Part of concerns from reviewers were addressed.

**Reviewer Scores:**

They may keep their initial scores.

---

### Decision · Program_Chairs · 2026-01-26

Reject